# Brain network dynamics during working memory are modulated by dopamine and diminished in schizophrenia

Urs Braun [1,2 ✉], Anais Harneit[1], Giulio Pergola[3], Tommaso Menara [4], Axel Schäfer[5,6], Richard F. Betzel [7], Zhenxiang Zang[1], Janina I. Schweiger[1], Xiaolong Zhang[1], Kristina Schwarz[1], Junfang Chen[1], Giuseppe Blasi[3], Alessandro Bertolino[3], Daniel Durstewitz[8], Fabio Pasqualetti [4], Emanuel Schwarz [1], Andreas Meyer-Lindenberg[1], Danielle S. Bassett [2,9,10,11,12,13,14] & Heike Tost [1,14]

Dynamical brain state transitions are critical for flexible working memory but the network mechanisms are incompletely understood. Here, we show that working memory performance entails brain-wide switching between activity states using a combination of functional magnetic resonance imaging in healthy controls and individuals with schizophrenia, pharmacological fMRI, genetic analyses and network control theory. The stability of states relates to dopamine D1 receptor gene expression while state transitions are influenced by D2 receptor expression and pharmacological modulation. Individuals with schizophrenia show altered network control properties, including a more diverse energy landscape and decreased stability of working memory representations. Our results demonstrate the relevance of dopamine signaling for the steering of whole-brain network dynamics during working memory and link these processes to schizophrenia pathophysiology.

[1] Department of Psychiatry and Psychotherapy, Central Institute of Mental Health, Medical Faculty Mannheim, University of Heidelberg, Mannheim, Germany. [2] Department of Bioengineering, University of Pennsylvania, Philadelphia, PA, USA. [3] Department of Basic Medical Science, Neuroscience, and Sense Organs, University of Bari Aldo Moro, Bari, Italy. [4] Mechanical Engineering Department, University of California at Riverside, Riverside, CA, USA. [5] Bender Institute of Neuroimaging, Justus Liebig University Giessen, Gießen, Germany. [6] Center for Mind, Brain and Behavior, University of Marburg and Justus Liebig University Giessen, Gießen, Germany. [7] Department of Psychological and Brain Sciences, Indiana University, Bloomington, IN, USA. [8] Department of Theoretical Neuroscience, Central Institute of Mental Health, Medical Faculty Mannheim, University of Heidelberg, Mannheim, Germany. [9] Department of Psychiatry, University of Pennsylvania, Philadelphia, USA. [10] Department of Neurology, University of Pennsylvania, Philadelphia, USA. [11] Department of Physics & Astronomy, University of Pennsylvania, Philadelphia, USA. [12] Department of Electrical & Systems Engineering, University of Pennsylvania, Philadelphia, USA. [13] The Santa Fe Institute, Santa Fe, NM, USA. [14] These authors contributed equally: Danielle S. Bassett, Heike Tost. ✉email: urs.braun@zi-mannheim.de

Working memory is an essential part of executive cognition depending on prefrontal neurons functionally modulated through dopamine D1 and D2 receptor activation[1–3]. The dual-state theory of prefrontal dopamine function links the differential activation of dopamine receptors to two discrete dynamical regimes: a D1-dominated state with a high energy barrier favoring robust maintenance of cognitive representations and a D2-dominated state with a flattened energy landscape enabling flexible switching between states[4]. Recent accounts extend the idea of dopamine's impact on working memory from a local prefrontal to a brain-wide network perspective[5–7], emphasizing the dual role of dopamine in regulating the complex interplay between striatal and prefrontal circuits critical for balancing the stability-flexibility tradeoff. Indeed, several lines of research support the notion that dopamine actions in frontal-parietal regions contribute to both maintaining cortical representations[8–10], and the flexible switching between different representations[2,11,12]. Notably, a large body of evidence further demonstrates that the latter process additionally involves striatal-cortical interactions[6,9], suggesting a gating function of the striatum for cortical memory representations[7]. These accounts highlight the contribution of widespread neural circuits and their regulation by dopamine to working memory.

Several lines of work support the idea that proper working memory execution requires the ordered transition through global brain states and the flexible reconfiguration of brain-wide interactions[13–15]. Notably, multiple studies have demonstrated that these temporal reconfigurations are altered in schizophrenia[13,16]. The complex dynamics of state transitions unfold upon the underlying structural scaffold whose architecture shapes the structure-function relationship[17,18] and constrains the dynamic repertoire enabling executive functioning[19–21]. However, it remains unclear how the brain controls, steers, and guides transitions between these different states and which brain components underlie the coordinated adaptation of brain-wide activity patterns.

A promising tool to study these questions and concepts derived from the dual-state theory is network control theory (NCT). NCT has been recently introduced to neuroscience[22,23] and can be used to model brain network dynamics as a function of interconnecting white matter tracts and regional control energy[23]. Such dynamics are framed upon brain states, which are defined as a whole-brain pattern of activity at a given moment in time. Based on the structural connectome, NCT can be used to examine the landscape of brain activity states: That is, which states would the system have difficulty accessing, and more importantly, which regions need to be influenced and to what extent to make those states accessible or to maintain those states[24].

Here, using NCT, we study transitions between (and the stability of) whole-brain neural states measured by fMRI during a well-established working memory task (Fig. 1). Building on a brain parcellation spanning both cortical and subcortical areas (see "Methods"), we define brain states as individual brain activity patterns related to a working memory condition (2-back) and to an attention control condition requiring motor response (0-back) in a sample of 178 healthy individuals undergoing fMRI. Separately, we obtain structural connectomes from fiber tracking along white matter tracts measured by diffusion tensor imaging (DTI) data acquired in the same participants. We approximate brain dynamics locally by a simple linear dynamical system and compute the local and global control energy necessary to drive or maintain certain activity patterns.

Within this statistical framework, we test the following predictions of the dual-state theory of brain network function and evaluate their implications for schizophrenia. Specifically, (I) we

posit that brain states related to high cognitive effort are harder to maintain, e.g., exhibit decreased stability and require more control energy to be accessed, than states related to low cognitive effort. (II) Using individual predicted gene co-expression indices for dopamine receptors, we test the hypothesis that the stability of brain states should be related to D1 receptor function, while D2 receptor function should be associated with decreased efforts to flexibly switch between global activity patterns. (III) We further validate these hypotheses and provide mechanistic support by pharmacologically blocking D2-receptor function in vivo, which should drive the system into a state where the switching between activity patterns requires more effort, e.g., demand more control energy. Finally, (IV) by considering the known cognitive deficits and structural network alterations in schizophrenia, we test our prediction that patients experience a reduced ability to control global reconfigurations of brain states.

## Results

**Brain state stability and control over brain state transitions in health.** We begin by asking how the brain transitions between different cognitive states during the performance of a well-established N-back working memory task. We define individual brain states as spatial patterns of β estimates associated with activity across brain regions of interest during a working memory condition (2-back) and during an attention control condition requiring motor response (0-back). It is important to note that our definition of brain states relates to the statistical spatial pattern of β estimates from a general linear model and does not reflect neuronal activity occurring en masse as, for example, in neurophysiological animal experiments. To quantify the energy efforts associated with a specific transition from an initial state $x_0$ to a target state $x_T$, we approximate brain dynamics locally by a simple linear dynamical system,

$$\dot{\mathbf{x}}(t) = \mathbf{A}\mathbf{x}(t) + \mathbf{B}\,\mathbf{u}(t) \tag{1}$$

where $\mathbf{x}(t)$ is the brain state of the system, $\mathbf{A}$ is a structural connectome inferred from DTI data, $\mathbf{u}(t)$ is the control input, and $\mathbf{B}$ is a matrix describing which regions are control nodes. After finding the optimal control input $\mathbf{u}$ that enables a transition, the control energy of each node is calculated as the squared integral over time of $\mathbf{u}$; intuitively, this quantity measures the control input that the node has to exhibit to facilitate the transitions from the initial state to the target state. Similarly, the stability of a brain state can be defined as the inverse control energy needed to maintain a specific state. In this framework, control energy can be interpreted as the effort of a brain region needed to steer the activity pattern of itself and its connected brain regions into the desired final activation state; relatedly, stability can be interpreted as the effort of a brain region needed to maintain a given activity pattern of itself and its connected brain regions. For a more detailed mathematical description of the network control framework, please see the "Methods" section and Supplemental Information.

Following the control theory framework, we started by computing the stability of both cognitive states as well as the control energy of the transitions between them in a sample of healthy individuals (Table 1). As expected, the cognitively more demanding 2-back brain state was less stable (i.e., required higher energy for maintenance) than the motor control state (Fig. 2a; repeated-measures ANOVA: main effect of 0- vs. 2-back stability: $F(1,173) = 66.80$, $p < 0.001$, age, sex and brain activity as covariates of non-interest). Further, the stability of the 2-back state was significantly associated with working memory accuracy (Fig. 2b; $b = 0.274$, $p = 0.006$, age, sex, and brain activity as covariates of non-interest). These findings suggest that more

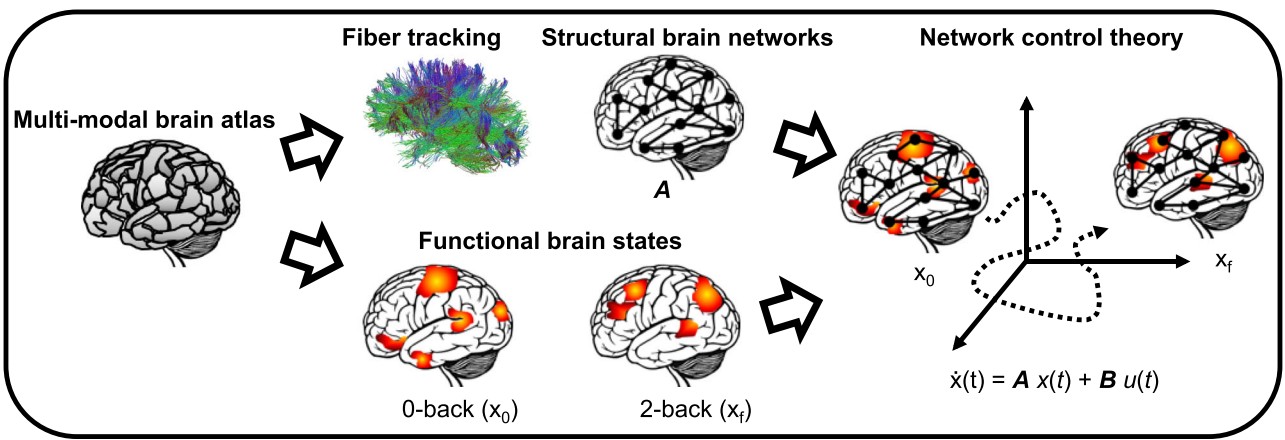

**Fig. 1 Theory and methods.** A summary of the methods to assess brain dynamics using network control theory. We use a multimodal atlas and apply it to both diffusion tensor imaging to obtain a structural connectome, and to functional magnetic resonance imaging to obtain activation patterns during 0-back and 2-back working memory tasks. Finally, we use network control theory to explain transitions between 0-back and 2-back states based on the underlying structural connectome. Here, $x_O$ denotes the initial state of the system, $x_f$ denotes the desired final state, $x(t)$ is the state of the system at time $t$, $A$ is the wiring diagram of the underlying network, $B$ denotes an input matrix defining the control nodes, and $u(t)$ is the time-dependent control signal. For further mathematical details, please see the "Methods" section.

| Table 1 Characteristics for the healthy control and schizophrenia samples. | | | | | |
|---|---|---|---|---|---|
| | **Healthy controls (n = 178)** | **Matched controls (n = 80)** | **Individuals with schizophrenia (n = 24)** | **t or $\chi^2$ value** | **P value** |
| *Demographic information* | | | | | |
| Age (year) | 33.05 ± 10.98 | 35.49 ± 10.55 | 32.25 ± 10.33 | 1.32 | 0.188 |
| Sex (male/female) | 93/85 | 46/34 | 18/6 | 2.39 | 0.122 |
| Years of education | 13.66 ± 2.41 | 13.65 ± 2.73 | 11.68 ± 1.45 | 2.72 | 0.008 |
| *Psychological assessments* | | | | | |
| MWTB | 30.74 ± 3.84 | 30.32 ± 4.83 | 29.13 ± 3.27 | 1.11 | 0.272 |
| PANSS positive | n.a | n.a. | 12.50 ± 6.76 | – | – |
| PANSS negative | n.a | n.a | 15.17 ± 6.76 | – | – |
| BDI | n.a | n.a. | 12.42 ± 7.71 | – | – |
| Years of illness | n.a. | n.a. | 10.22 ± 9.32 | – | – |
| *fMRI task performances* | | | | | |
| Accuracy (%) | 80.10 ± 18.35 | 68.75 ± 19.33 | 65.54 ± 19.79 | 0.70 | 0.479 |
| Reaction time (ms) | 496.15 ± 279.50 | 589.21 ± 286.80 | 627.06 ± 306.99 | −0.6 | 0.578 |
| *Head motion parameters* | | | | | |
| fMRI: Mean frame-wise displacement (mm) | 0.15 ± 0.06 | 0.20 ± 0.09 | 0.19 ± 0.08 | −1.11 | 0.270 |
| DTI: Mean absolute root-mean-square displacement (mm) | 1.27 ± 0.74 | 1.37 ± 0.89 | 1.34 ± 0.59 | 0.18 | 0.860 |
| DTI: tSNR | 5.63 ± 0.45 | 5.52 ± 0.49 | 5.26 ± 0.49 | 2.21 | 0.028 |

Source data are provided with this paper.
*MWTB* Mehrfach Wortschatz Intelligenztest B, a German multiple-choice vocabulary intelligence test as a measure of premorbid IQ, *PANSS* positive and negative symptom scale, *BDI* Beck's depression inventory, *DTI* diffusion tensor imaging, *tSNR* temporal signal-to-noise.

stable 2-back network representations in the form of whole-brain activity patterns[25] support better working memory performance.

We next studied how the brain flexibly changes its activity pattern between states. Transitioning into the cognitively more demanding 2-back state required more control energy than the opposite transition (Fig. 2c; repeated-measures ANOVA: $F(1174) = 27.98$, $p = 0.001$, age, sex and difference in brain activity as covariates of non-interest). To investigate which brain regions are the most important controllers in these transitions, we sought to quantify the influence that a single brain region has on the entire system's dynamics during state trajectories. To that purpose, we computed the control impact of each node by iteratively removing one brain region from the network and re-computing the change in control energy. Exploratory visualization of the 20% of brain regions that exhibited the highest control impact in both transitions suggested that prefrontal and parietal cortices steer both types of transitions, while default-mode areas are preferentially

important for switching to the more cognitively demanding state (Fig. 2d; see SI for illustration of alternative thresholds).

**Brain state stability and control energy relate to predicted frontal dopamine D1 and D2-receptor expression.** Following from the dual-state theory of network function, the stability of task-related brain states should be related to prefrontal D1 receptor status. To estimate individual prefrontal D1 receptor expression in each participant, we utilized methods relating prefrontal cortex D1 and D2 receptor expression to genetic variation in their co-expression partner, thereby enabling us to predict individual dopamine receptor expression levels from genotype data across the whole genome[26,27]. Specifically, previous work using weighted Gene Co-expression Network Analysis[28] applied on the Braincloud dataset of post-mortem DLPFC gene expression[29] had identified 67 non-overlapping sets of genes

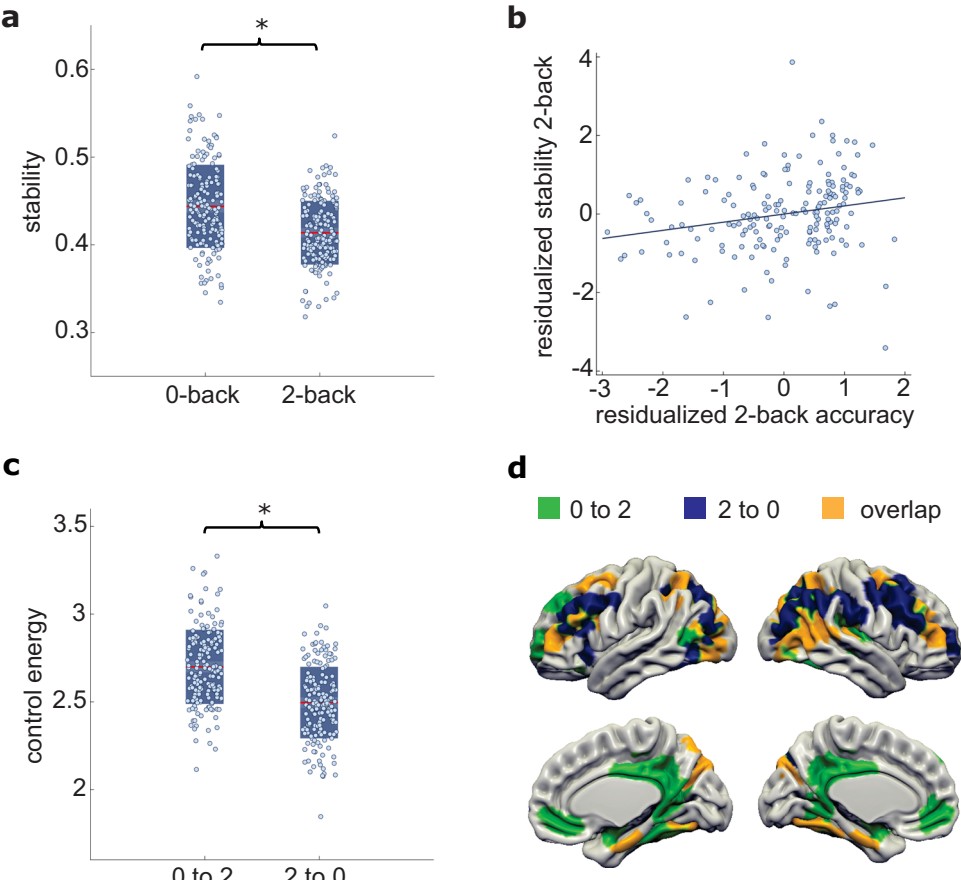

**Fig. 2 Controllability and stability of brain dynamics during working memory. a** The stability of the 2-back state reflecting working memory activity is lower than that of the 0-back state reflecting motor and basic attention control activity ($F(1173) = 66.80$, $p < 0.001$). Red lines indicate mean values and boxes indicate one standard deviation of the mean. **b** Associations of 2-back stability with working memory performance (accuracy: $b = 0.274$, $p = 0.006$; covarying for age, sex, and mean activity). **c** Steering brain dynamics from the control condition to the working memory condition (0–2) requires more control energy than vice versa ($F(1174) = 27.98$, $p < 0.001$). **d** Unique and common sets of brain regions contributing to the transition from 0-back to 2-back and the transition from 2-back to 0-back, respectively. For illustrative and exploratory purposes, we projected the computed control impact of each brain region for the respective transitions on a 3D structural template, displaying the regions with the 20% highest control impact for each transition (see SI for the illustration of alternative thresholds). Black lines indicate mean, dark boxes indicate 1 standard deviation, light boxes indicate 1.96 SEM and asterixis denote significance at $p < 0.05$. Source data are provided with this paper.

based on their expression pattern. The co-expression gene sets including DRD1 and DRD2 were summarized into Polygenic Co-expression Indices (PCIs) based on weighted SNPs that predicted co-expression of these genes. Based on these weighted sums of SNPs, we calculated individual PCI scores related to D1 and D2 receptor expression for a subset of 64 individuals for which human GWAS data were available (for more details see "Methods" and SI). In these individuals, we found that the D1 (but not D2) expression-related gene score predicted stability of both states (Fig. 3a; 0-back: $b = 0.184$, $p = 0.034$; 2-back: $b = 0.242$, $p = 0.007$, age, sex, brain activity, and first 5 genetic PCA components as covariates of non-interest), in line with the assumed role of dopamine D1-related signaling in maintaining stable activity patterns during task performance[4,8].

Following predictions from the dual-state theory, switching between different activity representations should relate to dopamine D2 receptor function, independent of stability. Indeed, when controlling for stability as a nuisance covariate in the regression model, the control energy of both state transitions could be predicted by the D2 (but not the D1) receptor expression gene score (Fig. 3b; 0- to 2-back: $b = -0.076$, $p = 0.037$; and trending for 2- to 0-back: $b = -0.134$, $p = 0.068$, age, sex, the difference in brain activity, and first 5 genetic PCA components

as covariates of non-interest), in line with the assumed role of dopamine D2 receptor function in lowering energy barriers between states[4] and thus enabling flexible switching between neural representations[8].

**Brain state transitions can be modulated by D2 receptor antagonism.** Our results thus far support the notion that the brain is a dynamical system in which the stability of a state is substantially defined by cognitive effort and modulated by D1 receptor expression, while transitions between states depend primarily on D2 receptor expression. If true, such a system should be sensitive to dopaminergic manipulation, and interference with D2-related signaling should reduce the brain's ability to control its optimal trajectories, i.e., increase the control energy needed when switching between states. To test these hypotheses, we investigated an independent sample of healthy controls ($n = 16$, Table 2) receiving 400 mg Amisulpride, a selective D2 receptor antagonist, in a randomized, placebo-controlled, double-blind pharmacological fMRI study. Importantly, because no DTI data were acquired in this study, we used a link-wise averaged connectivity matrix from all healthy subjects in study 1, following previous work[30,31].

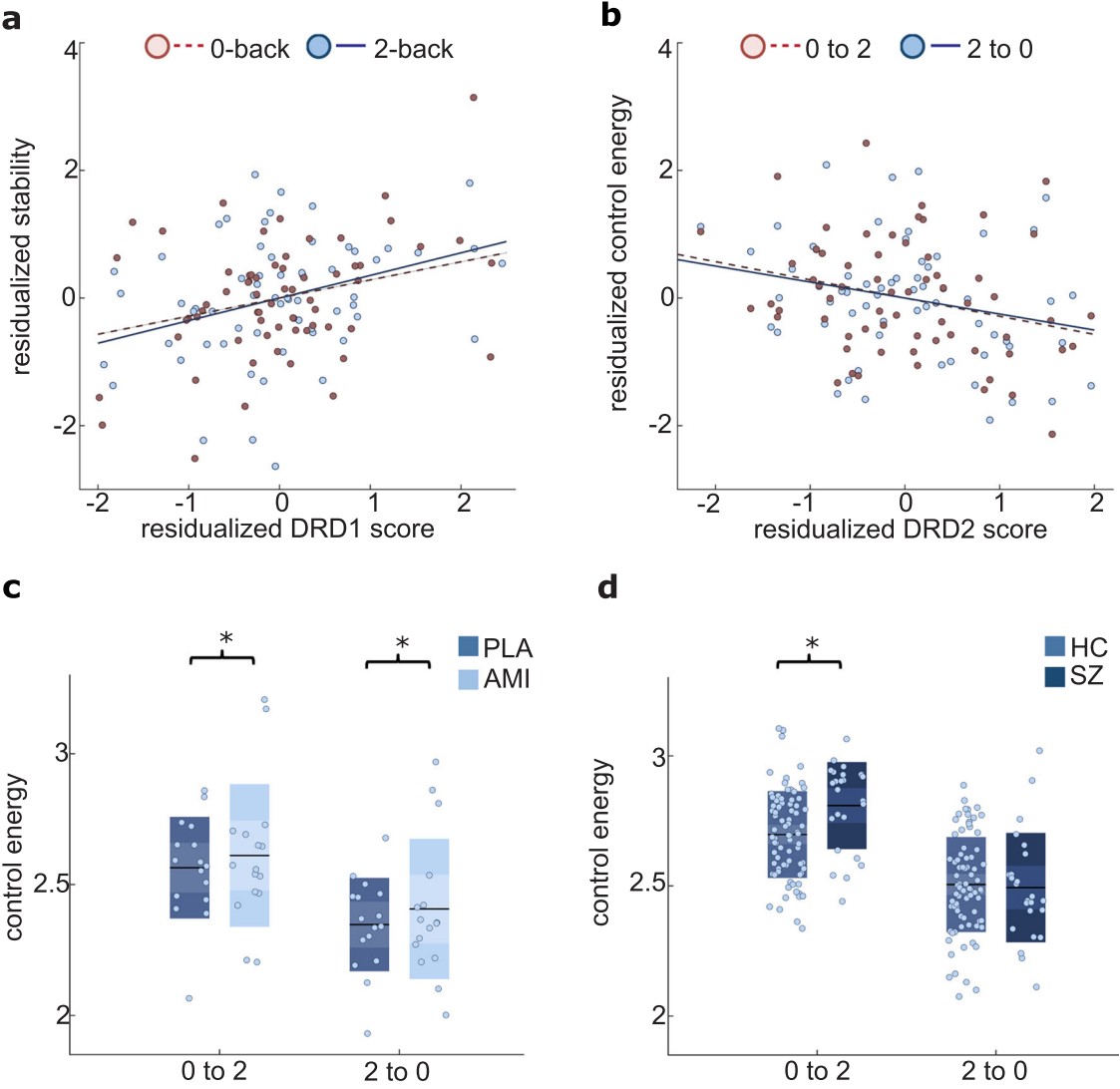

**Fig. 3 Dopamine receptor expression and pharmacological modulation impact whole-brain dynamics. a** Genetic scores predicting DRD1 expression in prefrontal regions positively predict stability of both brain states (0-back: $b = 0.184$, $p = 0.034$; 2-back: $b = 0.242$ $p = 0.007$; age, sex, mean brain state activity, and first 5 genetic PCA components as covariates of non-interest). **b** Genetic scores predicting DRD2 expression in prefrontal regions negatively predict control energy for both brain state transitions (0-back to 2-back: $b = -0.076$, $p = 0.037$; and trend wise for 2-back to 0-back: $b = -0.134$, $p = 0.068$; age, sex, mean brain activity difference, and first 5 genetic PCA components, stability of 0-back and 2-back as covariates of non-interest). **c** Amisulpride (AMI) increases control energy for transitions in comparison to placebo (PLA) (main effect of drug: $F(1,10) = 7.27$, $p = 0.022$; interaction drug by condition: $F(1,10) = 0.42$, $p = 0.665$, activity difference, drug order, and sex as covariates of non-interest). Black lines indicate mean values and boxes indicate one standard deviation of the mean. **d** Individuals with schizophrenia (SZ) need more control energy when transitioning into the working memory condition than matched healthy controls (HC) ($F(1,98) = 5.238$, $p = 0.024$, age, sex, tSNR and mean activity as covariates of non-interest), but not vice versa. Black lines indicate mean, dark boxes indicate 1 standard deviation, light boxes indicate 1.96 SEM. Source data are provided with this paper.

Consistent with our expectations, we observed that greater control energy was needed for transitions under D2 receptor blockade (Fig. 3c; repeated-measures ANOVA with drug and transition as within-subject factors; main effect of the drug: $F(1,10) = 7.27$, $p = 0.022$; drug-by-condition interaction: $F(1,10) = 0.42$, $p = 0.665$, sex, the difference in brain activity, and drug order as covariates of non-interest). In contrast, we observed no effect on the stability of states; that is, the inverse control energy required to stabilize a current state was not impacted by the pharmacological manipulation (main effect of drug: $F(1,8) = 0.715$, $p = 0.422$, sex, brain activity, and drug order as covariates of non-interest). Importantly, we were able to trend-wise replicate the results of the pharmacological intervention with risperidone, a substance showing lower D2-receptor selectivity (repeated-measures ANOVA with drug and transition as within-

subject factors, the main effect of the drug: $F(1,10) = 3.490$, sex, the difference in brain activity, and drug order as covariates of non-interest, $p = 0.091$; repeated-measures ANOVA with drug and stability as within-subject factors, the main effect of the drug: $F(1,8) = 1.057$, $p = 0.334$, sex, brain activity, and drug order as covariates of non-interest, see Supplementary Table 1).

**Reduced brain state stability and control over brain state transitions in schizophrenia.** Dopamine dysfunction, working memory deficits, and alterations in brain network organization are hallmarks of schizophrenia[13,32–34]. We therefore tested for differences in the state stability and in the ability to control state transitions between 24 individuals with schizophrenia and a healthy control sample balanced for age, sex, performance, head

**Table 2 Sample characteristics for the pharmacological intervention study.**

| | Healthy control ($n = 16$) | | t value | P value |
|---|---|---|---|---|
| | Placebo | Amisulpride | | |
| *Demographic information* | | | | |
| Age (year) | 26.63 ± 5.34 | | | |
| Sex (male/female) | 8/8 | | | |
| *fMRI task performances* | | | | |
| Accuracy (%) | 89.32 ± 10.00 | 85.68 ± 11.66 | 1.27 | 0.223 |
| Reaction time (ms) | 347.42 ± 104.26 | 365.40 ± 142.54 | −0.582 | 0.569 |
| *Head motion parameters* | | | | |
| fMRI: Mean frame-wise displacement (mm) | 0.124 ± 0.03 | 0.126 ± 0.04 | −0.301 | 0.767 |

Source data are provided with this paper.

motion, and premorbid IQ, but different in signal-noise-ratio (tSNR) (Table 1). Stability in individuals with schizophrenia was reduced for the cognitively demanding working memory state ($F(1,98) = 6.43$, $p = 0.013$, age, sex, brain activity, and tSNR as covariates of non-interest), but not for the control condition ($F(1,98) = 0.052$, $p = 0.840$, same covariates; Supplementary Table 1). The control energy needed for the 0- to 2-back transition was significantly higher in schizophrenia than in controls (Fig. 3d; $F(1,98) = 5.238$, $p = 0.024$, age, sex, the difference in brain activity, and tSNR as covariates of non-interest), while the opposite transition (2-back to 0-back) showed no significant group difference (ANOVA: $F(1,98) = 0.620$, $p = 0.433$, same covariates; Supplementary Table 1). These results suggest that the brain energy landscape is more diverse in schizophrenia than in controls, making the system more difficult to steer appropriately.

To further strengthen this notion, we estimated the variability in suboptimal (higher energy) trajectories connecting different cognitive states by enacting subtle random perturbations to the minimum energy trajectories over 200 iterations (see "Methods"). In a diversified energy landscape, we expected that the variation of trajectories around the minimum-energy trajectory should be larger than in the less diversified energy landscape of healthy controls, implying that small perturbations may have a more substantial impact in schizophrenia. In line with our hypothesis, we found that the variability in such perturbed trajectories was indeed increased in schizophrenia compared to controls (rm-ANOVA: main effect of group: $F(1,98) = 4.789$, $p = 0.031$, age, sex, tSNR as covariates of non-interest). Together, these findings suggest that working memory brain states are less stable in schizophrenia, harder to control, and more susceptible to disturbance compared to healthy controls.

## Discussion

Working memory is a core cognitive function that encompasses the ability to maintain but also update and manipulate cognitive representations to successfully perform ongoing tasks. The underlying neural substrate of these processes critically depends on the coordinated reconfiguration and persistence of distinct global neural activity patterns that are modulated by the differential actions of dopamine signaling. In the present study, we provide evidence that the stability of and switching between global brain activation patterns during working memory can be meaningfully assessed by NCT, providing a putative mechanism for the brain's capacity to control the unfolding of activity patterns on top of the underlying structural connectome. We further demonstrate that these processes are modulated by dopamine signaling in accordance with the dual-state theory of dopamine, and that they are altered in schizophrenia as expected by current pathophysiological network hypotheses[35–37] in this disorder.

Our study extends the current knowledge in the field in several notable ways. Firstly, our results demonstrate that the cognitively more demanding state exhibits lower stability, i.e., requires more control energy to be maintained. Indeed, tasks requiring additional working memory demands have been shown to be associated with increased metabolic and attentional demands[38], as well as exhibiting more extended spatial distributions of activity[39,40] and less neuronal selectivity[41], which could account for the increased control efforts needed to actively retain and steer the neural system towards this dynamic network state. Notably, we also found that the stability of the cognitively demanding state was further associated with better working memory performance, complementing previous evidence demonstrating that increased brain state stability is accompanied by greater behavioral performance across a spectrum of tasks[42,43].

Further, switching to the cognitively more demanding and less stable state required more control energy than the inverse transition. The direction of these differences suggests that the cognitively more demanding state was more difficult to access, a notion that is in line with the idea of associated switch costs when turning to more difficult tasks[44]. Importantly, these analyses were performed while controlling for the effects of mean brain activation; therefore, the results cannot be explained as mere epiphonema of global or local brain activity levels as measured by fMRI, which have previously been shown to be task- and cognitive-load dependent[38,45]. When examining the regional contributions of brain areas to the control of state switching, we are able to differentiate between (i) a universal set of brain regions mainly located in prefrontal and parietal cortices supporting both transitions, and (ii) medial structures in default-mode related areas that showed particularly strong involvement in 0- to 2-back transitions. These results are in line with previous predictions from network control studies indicating that PFC areas are essential for controlling transitions into hard-to-reach states[22,46], and further support the assumed role of frontal-parietal circuits in steering brain dynamics[47] and their prominent role in shifting tasks[9,48–50]. The findings also emphasize the importance of the coordinated behavior of brain systems commonly displaying deactivations during demanding cognitive tasks[51].

Secondly, in line with the prediction of the dual-state theory of network function, we show that the ability to control brain dynamics during working memory is differentially modulated by D1/D2 dopamine receptor functioning. D1-receptor signaling in frontal circuits has been previously shown to facilitate working memory by tuning signal-to-noise ratios in pyramidal neurons[1,52], enabling stable network activation patterns that support the maintenance of neural representations[4,53]. In contrast, D2-receptor activation in PFC can lead to decreased GABA and NMDA receptor-related currents, thereby counteracting D1-receptor activity and ultimately enabling higher flexibility and switching between cognitive representations[4]. While these insights focus on cortical microcircuits and are derived from animal studies and theoretical modeling, our results complement previous work demonstrating that similar principles govern the modulatory actions of D1 (and D2 receptors) at the macroscopic level of brain-wide networks, particularly in frontal-parietal circuits[6].

It is important to note that our data provide an association of whole-brain processes with PCIs derived from prefrontal areas, and that previous studies have demonstrated differential expression patterns of dopamine receptors for prefrontal and striatal areas as well as differential (and even antagonist) behavior of dopamine receptor stimulation in striatal and prefrontal circuits[6,54]. Therefore, it seems plausible that our results mainly reflect dopamine actions as observed in PFC-related circuits, which are also the dominant control nodes in our model

facilitating both state transitions. Both observations would support a model of frontal-parietal circuits serving as hub regions modulated by D2-receptor signaling, which controls and facilitates the flexible adaptation of brain-wide activity and connectivity patterns[14,48,49,55,56]. While our model concentrates on PFC-related dopamine action, it does not exclude the increasingly important concept that emphasizes the additional role of striatal input and output gating as dopamine-related mechanisms contributing to a stability-flexibility tradeoff critical for cognitive control, task-switching, and working memory[7,57,58]. However, studying the differential contributions of striatal and frontal dopamine signaling on working memory in future studies will require a finer-grained task design to disentangle the several cognitive subprocesses that are currently mingled in the two conditions of our N-back task. The idea of dopamine-related frontal-parietal circuits as important regions for flexibly controlling the reconfiguration of brain-wide activity patterns is further supported by our pharmacological intervention, where we observe a specific increase of control energy for switching brain activation patterns after D2-blockade, but no effect on the stability of these patterns. Here, future work quantifying brain-wide D1- and D2-receptor levels in vivo in combination with pharmacological manipulations would provide valuable and strongly needed data to disentangle the specific contributions of the spatial distribution and different receptor subtypes to working memory processes.

Thirdly, we show that individuals with schizophrenia show reduced controllability and stability of working memory network dynamics, consistent with the idea of an altered functional architecture and energy landscape of cognitive brain networks. Cognitive deficits in schizophrenia have been repeatedly linked to altered dopamine-related brain activations in prefrontal circuits, showing a complex U-shaped dependency shaped by genetics, task difficulty, medication, and dopamine levels[3,13,33,56,59–62]. In previous work we have shown that these deficits are accompanied by less stable, brain-wide network configurations during working memory[13], extended by our current observations that the stability of brain activation patterns is specific to cognitive demanding states. Importantly, our samples were balanced in terms of working memory performance, and therefore the observed differences are unlikely a mere result of performance levels.

Although less stable, cognitive states were more difficult to access, which seemingly contradicts our results of increased control energy after D2-blockade, but is in line with clinical observations that D2-blockade does not ameliorate cognitive symptoms in schizophrenia[63]. Instead, our additional analysis on the variability of trajectories suggests that the increased demand for control energy in our individuals with schizophrenia is potentially a direct result of a more diverse and therefore harder-to-navigate control landscape. This landscape is mainly shaped by the topology of the underlying structural connectome[64], for which meso-scale alterations, such as reduced modularity and disruptions of the rich-club[65–67], are prominent in schizophrenia. Such alterations are thought to result in a biased trade-off between integration and segregation[68–70], and in an increased subtle randomization[71]. Our results therefore provide a mechanistic explanation for how connectome disruption can lead to altered brain dynamics and cognition in schizophrenia.

Several aspects of our work require special consideration. Firstly, to relate brain dynamics to cognitive function, we focus on discrete "meta-level" brain states where each state is summarized by a single brain activation pattern rather than a linear combination of multiple brain activity patterns. These brain states do not purely reflect a single process but instead involve several cognitive subprocesses. Future studies could use specific paradigms to disentangle these subprocesses, in combination with

more direct and higher time-resolved measures of brain activity such as MEG. Secondly, although we could demonstrate a link between brain dynamics, measured by means of control energy, and predicted prefrontal dopamine receptor expression, the link is indirect and requires confirmation by direct measurements. Thirdly, we cannot exclude the possibility that disorder severity, duration, symptoms, or medication may have influenced network dynamics in individuals with schizophrenia (see SI). Finally, while the sample sizes of our pharmacological and patient study are rather small, we were able to show comparable effects of dopaminergic manipulation on control properties using a second drug (see SI), further supporting the validity of the underlying rationale.

In summary, our data demonstrate the utility of NCT for the non-invasive investigation of the mechanistic underpinnings of (altered) brain states and their transitions during cognition. Our data suggest that engagement of working memory involves brain-wide switching between activity states and that the steering of these network dynamics is differentially, but cooperatively, influenced by dopamine D1- and D2-receptor function.

## Methods

**Participants and study design**. All participants provided written informed consent for protocols approved by the Medical Ethics Committee II of the Medical Faculty Mannheim at the Ruprecht-Karls-University in Heidelberg, Germany. For the first study including healthy controls and patients with schizophrenia, we included a total of 202 subjects (178 healthy controls, 24 individuals with schizophrenia, Table 1). A trained psychiatrist or psychologist verified the diagnosis of schizophrenia based on ICD−10 criteria.

For the second, pharmacological intervention study (Registration number: DRKS00005267 in the German Registry for Clinical Studies), 17 healthy individuals completed a subject- and observer-blind, placebo-controlled, randomized three-period cross-over study (Table 2). Participants were invited for a fixed interval of 7 days with each scanning session taking place at approximately the same time of day. On each of three scanning visits, individuals either received a single oral dose of 400 mg Amisulpride, 3 mg Risperidone, or Placebo. MRI scanning took place 2 h after drug administration, with the N-back paradigm commencing ~10 min after the start of the scan. One subject was excluded from the analysis due to an excessive body-mass index (BMI > 30).

## Data acquisition

*fMRI*. For the first study in healthy controls and individuals with schizophrenia, BOLD fMRI was performed on a 3T Siemens Trio (Erlangen, Germany) in Mannheim, Germany. Prior to the acquisition of functional images, a high-resolution T1-weighted 3D MRI sequence was conducted (MPRAGE, slice thickness = 1.0 mm, FoV = 256 mm, TR = 1570 ms, TE = 2.75 ms, TI = 800 ms, $\alpha$ = 15°). Subsequently, functional data were acquired during performance of the N-back paradigm (for details, see supplemental material) using an echo-planar imaging (EPI) sequence with the following scanning parameters: TR/TE = 2000/30 ms, $\alpha$ = 80°, 28 axial slices (slice-thickness = 4 mm + 1 mm gap), descending acquisition, FoV = 192 mm, acquisition matrix = 64 × 64 128 volumes. The visual stimuli of the N-back paradigm were presented using Presentation (https://www.neurobs.com).

For the pharmacological intervention study, BOLD fMRI was performed on a 3T Siemens Trio Scanner (Erlangen, Germany) using an echo-planar imaging (EPI) sequence with the following parameters: TR = 1790 ms, TE = 28 ms, 34 axial slices per volume, voxel size = 3 × 3 × 3 mm, 1 mm gap, 192 × 192 mm field of view, 76° flip angle, descending acquisition. Since the focus of this paper was on dynamical brain state transitions in the context of dopamine and working memory performance, other fMRI paradigms acquired in the pharmacological study are not reported here. These other fMRI paradigms include an emotional face-matching paradigm (Hariri task), a reward anticipation paradigm (monetary incentive delay task), and a resting-state scan. These data were not analyzed here as the topic of this work was restricted to the study of brain network dynamics during working memory processing and the data from these tasks are not suitable for testing hypotheses pertaining to the working memory domain. Outcome parameters on working memory-related brain activity are reported in the supplemental materials. The pharmacological challenge study with amisulpride and risperidone is registered with the German Clinical Trials Register (DRKS) under the registration number DRKS00005267. There are currently no plans for reporting other outcome measures of this trial.

*Diffusion tensor imaging (DTI)*. DTI data were acquired by using spin-echo EPI sequences with the following parameters: TR 14000 ms, TE 86 ms, 2 mm slice

thickness, 60 non-collinear directions, b-value 1000 s/mm2, 1 b0 image, FOV 256 mm.

**Connectome construction.** For the DTI data, the following preprocessing steps were performed with standard routines implemented in the software package FSL[72]: (i) correction of the diffusion images for head motion and eddy currents by affine registration to a reference (b0) image, (ii) extraction of non-brain tissues[73], and iii) linear diffusion tensor fitting. After estimation of the diffusion tensor, we performed deterministic whole-brain fiber tracking as implemented in DSI Studio using a modified FACT algorithm[74]. For each subject, 1,000,000 streamlines were initiated. Termination criteria included a maximum angle of 75 degrees and a minimal streamline length of 10 mm. We used a step size of 1 mm and did not apply smoothing. Streamlines with a length of <10 mm were removed[75]. For the construction of structural connectivity matrices, the brain was parcellated into 374 regions (see Supplemental Information). To map these parcellations into subject space, we applied a nonlinear registration implemented in DSI Studio. We estimated the structural connectivity between any two regions of the atlas by using the mean fractional anisotropy values between respective brain regions. This procedure resulted in a weighted adjacency matrix $\mathbf{A}$ whose entries $\mathbf{A}_{ij}$ reflect the strength of structural connectivity between two brain regions.

Because DTI data was not acquired during the pharmacological intervention study, we used the link-wise averaged connectivity matrix across all healthy subjects from study 1 to model the transition between individual brain states in the pharmacological intervention study[30,31]. A comparison of the structural connectomes between groups can be found in Supplementary Table 2.

**Brain state definition.** Because we were interested in investigating how the brain controls and transitions between global brain states underlying circumscribed cognitive processes (such as those supporting working memory, attention, and motor behavior), we defined brain states as stationary patterns of activity during the execution of these processes. It is important to note that the temporal resolution of fMRI and the design of the N-back task limit the investigation of differential cognitive processes contributing to memory performance on fast time scales. Therefore, we cannot investigate the detailed temporal dynamics of working memory processes. However, our simplified design allows us to extract meaningful brain states elicited by a controlled cognitive process and therefore enables us to relate brain dynamics to cognitive function. Specifically, we defined individual brain states as spatial patterns of beta estimates associated with activity across brain regions of interest during both conditions of the N-back task[76]. For that purpose, after standard preprocessing procedures in SPM12/8 (including realignment to the mean image, slice timing correction, spatial normalization into standard stereotactic space, resampling to 3 mm isotropic voxels, and smoothing with an 8 mm full-width at half-maximum Gaussian Kernel)[13,14,77], we estimated standard first-level general linear models for the N-back task, separately for each individual. Except for the SPM version, both studies followed the same preprocessing procedure. These GLM models included regressors for the 0-back and 2-back conditions of interest, as well as the 6 motion parameters as regressors of non-interest. To define the brain activity pattern associated with each condition of the task, we extracted GLM (beta) parameter estimates for the 0-back and 2-back conditions separately[76] and we averaged them across all voxels in each of the 374 regions without applying any threshold. This procedure yielded a 374 by 1 vector for each condition ($\mathbf{x}_{0\text{-back}}$, $\mathbf{x}_{2\text{-back}}$) per subject representing how strongly the BOLD response in each brain region was associated with the working memory (2-back) or the motor control condition (0-back), respectively. These vectors ($\mathbf{x}_{0\text{-back}}$, $\mathbf{x}_{2\text{-back}}$) defined the final and target brain states for the following network control analyses.

**Network control theory**
*Optimal control theory framework.* To model the transition between 0-back and 2-back brain states, we used the framework of optimal control, following prior work[23,24,78] implemented in MATLAB. Based on individual brain states $\mathbf{X} = [\mathbf{x}_1, \dots \mathbf{x}_n]$ (in our case simplified to $n = 2$ states: 0-back and 2-back, see above) and a structural brain network $\mathbf{A}$ for each subject, we approximated the local brain dynamics by a linear continuous-time equation,

$$\dot{\mathbf{x}}(t) = \mathbf{A}\mathbf{x}(t) + \mathbf{B}\,\mathbf{u}(t) \tag{2}$$

To model the flow among task-related brain activity states. In the model, $\mathbf{x}(t)$ is the state of the system at time $t$, $\mathbf{A}$ is the wiring diagram of the underlying network, $\mathbf{B}$ denotes an input matrix defining the control nodes, and $\mathbf{u}(t)$ is the time-dependent control signal[23,43]. Note that while the initial state ($\mathbf{x}_0$) and the target state ($\mathbf{x}_T$) are empirically defined, any states of the system at other times are virtual intermediate steps in the trajectories of the state-space model. In that state-space, we aim to identify a trajectory between state $\mathbf{x}_0$ and state $\mathbf{x}_T$ that is minimal in terms of the necessary control input signals as well as the distance of the trajectory. This choice is motivated by two complementary ideas: first that the brain minimizes its energy expenditure to perform that transition and second, that optimal transitions between states should be non-random walk in state space. These notions can be formalized by defining an optimization problem that minimizes a given cost function. We define this cost function by the weighted sum of the energy cost of the transition and the integrated squared distance between the transition states and the

target state. The problem of finding an optimal control input $\mathbf{u}^*$ that induces a trajectory from an initial state $\mathbf{x}_0$ to a target state $\mathbf{x}_T$ reduces to the problem of finding an optimal solution to the minimization problem of the corresponding Hamiltonian[24]:

$$\min[H(\mathbf{p}, \mathbf{x}, \mathbf{u}, t) = \mathbf{x}^{\mathsf{T}}\mathbf{x} + \rho\mathbf{u}^{\mathsf{T}}\mathbf{u} + \mathbf{p}^{\mathsf{T}}(\mathbf{A}\mathbf{x} + \mathbf{B}\mathbf{u})] \tag{3}$$

The parameter $\rho$ in this equation allows to penalize the energy used by the optimal input in relation to the deviation from the optimal trajectory when solving for the optimal control. As we had no specific hypothesis that either of these elements of the cost function should prevail, we used the default of $\rho = 1$.

By setting the input matrix $\mathbf{B} = \mathbf{I}_{N \times N}$, the identity matrix, we allow all brain regions to be independent controllers[43,78]. This is motivated by our analysis question to use a system-level and data-driven approach to identify regions contributing most to the transitions in an unbiased way in line with previous work[79–81]. A description and discussion of the parameter choices, as well as additional analyses and discussion on the relation of control energy and fMRI activity measures can be found in the supplemental material.

*Control energy, stability, and impact.* Control energy for each node $k_i$, $i = 1, \dots,$ m (m = total number of brain nodes), was defined as

$$E_{k_i} = \int_{t=0}^{T} \|\mathbf{u}^*_{k_i}(t)\|^2 \mathrm{d}t \tag{4}$$

i.e., the squared integral over time of energy input that the node has to exhibit to facilitate the transitions from the initial state to the target state[24,43,78]. While the neurobiological foundations of control properties in the brain are not yet well understood, and our framework for defining control energy is based on a linear systems approach and cannot be directly related to the physical definition of the word energy (with joule as derived unit), this definition can be interpreted as the effort of a brain region needed to steer the activity pattern of itself and its connected brain regions into the desired final activation state, for example by tuning their internal firing or activity patterns by recurrent inhibitory connections. Accordingly, the total control energy for the entire brain was defined as the sum of all control energy across all nodes

$$E = \sum_{i=1}^{m} E_{k_i} \tag{5}$$

yielding one value for each transition per subject. To ensure a normal distribution of metric values for subsequent statistical testing, we applied a logarithmic transformation (base 10) to the control energy[78].

From the control energy, we can also obtain the control stability and control impact. Stability was defined as

$$S = \frac{1}{\log_{10}(E_{x_o = x_T})} \tag{6}$$

i.e. the inverse control energy needed to *maintain* a state, or in other words, the control energy needed to go, e.g., from 2-back to 2-back[43]. The rationale here is, that the energy required to maintain a state is inversely related to the distance of the state from a local minimum on the energy landscape. States that are distant from a local minimum will also dissipate quickly under spontaneous activity, while states at the local minimum will not. Thus, the temporal stability of a state is inversely related to the control energy required to maintain a state. For further details, we refer the interested reader to refs. [23,80,82,83]. To further investigate the influence that a single brain region has on the entire system's dynamics during state trajectories, we computed the control impact of each node by iteratively removing one brain region from the network and re-computing the change in control energy[24].

*Suboptimal trajectories.* To investigate the energy landscape surrounding the minimum energy (in this sense 'optimal') trajectories, we quantified the variability of suboptimal trajectories by adding subtle random perturbation to the minimum energy trajectories over 200 iterations. Note that we employed a discrete-time dynamical system rather than a continuous-time system for these analyses, as discrete-time systems are computationally more tractable. To discretize our linear continuous-time system, we employed the following transformations

$$\bar{\mathbf{A}} \triangleq e^{\mathbf{A}T_s} \tag{7}$$

and

$$\bar{\mathbf{B}} \triangleq \left( \int_0^{T_s} e^{\mathbf{A}(T_s - \tau)} d\tau \right) \mathbf{B} \tag{8}$$

where $\bar{\mathbf{A}}$ and $\bar{\mathbf{B}}$ are the corresponding structural matrix and control input matrix in a discrete time system and $T_s$ is the sampling time. As there is no prescriptive way to choose $T_s$, we estimated $T_s = \frac{1}{10}RT$, where $RT$ is the rise time of its fastest mode, i.e., the time that the system requires to go from 10 to 90% of its fastest step response. In the resulting discrete-time dynamical system

$$\bar{\mathbf{x}}(t + 1) = \bar{\mathbf{A}}\bar{\mathbf{x}}(t) + \bar{\mathbf{B}}\bar{\mathbf{u}}(t) \tag{9}$$

at each discrete time step $t$, we applied a principal component analysis (PCA) to the cloud of suboptimal points to reduce dimensionality. Because the previous

simulations show that the first PCA component explains more than 90% of variance, we continued by using the first component as a summary measure of suboptimal trajectories. To estimate the relative distance between suboptimal and optimal trajectories, we computed the percentage of variation of the maximum distance of the projected suboptimal points on the first principal component from the optimal one. This procedure resulted in a normalized measure giving the percentage of deviations for the 200 suboptimal trajectories from the optimal trajectory. Due to the heuristic nature of the algorithm applying a random perturbation, results can vary from run to run. Therefore, to increase replicability of our results, we repeated our analysis 10 times per subject ($10 \times 200 = 2000$ suboptimal trajectories per subject).

### Gene-based PCIs

*Genotyping, imputation, and quality control.* In this study, we used human genome data of 63 healthy subjects who were genotyped using HumanHap 610 and 660w Quad BeadChips. For all subjects, standard quality control (QC) and imputation were performed using the Gimpute pipeline (Chen, Lippold et al. 2018)[84] and the following established QC steps were applied. Step 1: Determine the number of male subjects for each heterozygous SNP of chromosome 23 and remove SNPs whose number is larger than 5% of the number of male samples. Step 2: Determine the number of heterozygous SNPs on X chromosome for each male sample, and remove samples that have the number of heterozygous SNPs larger than 10. Step 3: Remove SNPs with missing genotyping rate >5% before sample removal. Step 4: Exclude samples with missingness ≥ 0.02. Step 5: Exclude samples with autosomal heterozygosity deviation |Fhet| ≥ 0.2. Step 6: Remove SNPs with the proportion of missing genotyping >2% after sample removal. Step 7: Remove SNPs if the Hardy-Weinberg equilibrium exact test $P$-value was $<1 \times 10^{-6}$. Step 8: PCA was applied to detect population outliers. Imputation was carried out using IMPUTE2/ SHAPEIT[85–87], with a European reference panel for each study sample in each 3 Mb segment of the genome. This imputation reference set is from the full 1000 Genome Project dataset (August 2012, 30,069,288 variants, release "v3.macGT1"). The length of the buffer region is set to be 500 kb on either side of each segment. All other parameters were set to default values implemented in IMPUTE2. After imputation, SNPs with high imputation quality (INFO ≥ 0.6) and successfully imputed in ≥20 samples were retained. From the final well-imputed dataset with 63 subjects, we extracted 8 SNPs for DRD2 and 13 SNPs for DRD1[26,27,88]. For the subsequent genetic imaging analyses, we only used 64 subjects for whom all data modalities (DTI, fMRI, and imputed genome-wide SNP data) were available.

*Polygenic co-expression index calculation.* Recent publications have shown that gene sets defined using co-expression networks and selected for their association with the genes DRD1 and DRD2 provided replicable predictions of n-back-related brain activity and behavioral indices in line with the role of prefrontal dopamine in working memory[27,88–90]. The gene sets have been previously identified using weighted Gene Co-expression Network Analysis [WGCNA][28] applied on the Braincloud dataset ($N = 199$) of post-mortem DLPFC gene expression[29] resulting in 67 non-overlapping sets of genes based on their expression pattern. The co-expression gene sets including DRD1 and DRD2 were summarized into PCIs based on SNPs that predicted co-expression of these genes (called co-expression quantitative trait loci, or co-eQTLs). For the current analysis, we extracted the 13 SNPs linked to DRD1 and 8 SNPs linked to DRD2 expression as previously identified[26,27,88], and computed a weighted sum for both receptors similar to a genetic risk score computation for each individual. These PCIs scores served as a proxy for D1/D2 receptor expression in the DLPFC.

For further details please see supplemental information.

### Statistical inference

Statistical inference was performed using the Statistical Package for the Social Sciences 24 (IBM SPSS Statistics for Windows, Version 24) and R (https://www.Rproject.org/). All statistical comparisons were performed while controlling for age and sex and were tested two-sided. Because we were interested in control properties of brain state transitions independent of differences in the amount of mere activation, we controlled for the respective parameters reflecting the individual differences in activations in all analyses involving control properties. In particular, for all analyses involving stability measures, we additionally controlled for the average brain activity defined as the GLM parameter estimates over all regions in the 0-back and 2-back conditions. As the control energy of a transition depends highly on the absolute difference in the magnitude of the brain activity between its initial state and its final state (i.e., in our case the brain-wide activation difference between both task conditions) and as we were interested in the unique control properties independent of traditional activation differences, we additionally controlled for the difference in the mean brain activity in each analysis involving control energy. The difference in the mean brain activity was computed as the absolute average difference in the unthresholded GLM parameter estimates over all regions between 0-back and 2-back conditions. In all analyses involving polygenic scores for D1-/D2-expression, we further used the first 5 principal components from the PCA on the linkage-disequilibrium pruned set of autosomal SNPs to control for population stratification. Differences between control energy/stability of both conditions were assessed using repeated-measures ANOVA with the condition as a within-subject factor. Drug effects were modeled using a repeated-measures ANOVA with drug and condition as within-subject

factors. As healthy control and individuals with schizophrenia differed in one of the quantified DTI imaging quality parameters (tSNR, see Table 1), we also controlled for tSNR in each analysis involving both groups.

**Reporting summary**. Further information on research design is available in the Nature Research Reporting Summary linked to this article.

## Data availability

Data supporting the findings of this study are available upon reasonable request from the corresponding author. Data sharing is subject to GDPR restrictions. Raw data containing sensible information that can be used to identify individuals, such a whole-genome data, cannot be shared under current data protection laws. Source data are provided with this paper.

## Code availability

Associated Code can be found here: https://github.com/search?q=network_control_and_ dopamine Source data are provided with this paper.

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

## Acknowledgements

The authors thank all individuals who have supported our work by participating in our studies. There was no involvement by the funding bodies at any stage of the study. We thank Oliver Grimm, Leila Haddad, Michael Schneider, Natalie Hess, Sarah Plier, and Petya Vicheva for valuable research assistance. The authors thank Jason Kim and Lorenzo Caciagli for valuable feedback on the manuscript. U.B. acknowledges grant support by the German Research Foundation (DFG, grant BR 5951/1-1). H.T. acknowledges grant support by the DFG (Collaborative Research Center SFB 1158 subproject B04, Collaborative Research Center TRR 265 subproject A04, GRK 2350 project B2, grant TO 539/3−1) and German Federal Ministry of Education and Research (BMBF, grants 01EF1803A project WP3, 01GQ1102). A.M.L. acknowledges grant support by the DFG (Collaborative Research Center SFB 1158 subproject B09, Collaborative Research Center TRR 265 subproject S02, grant ME 1591/4−1) and BMBF (grants 01EF1803A, 01ZX1314G, 01GQ1003B), European Union's Seventh Framework Programme (FP7, grants 602450, 602805, 115300 and HEALTH-F2-2010-241909, Innovative Medicines Initiative Joint Undertaking (IMI, grant 115008) and Ministry of Science, Research and the Arts of the State of Baden-Wuerttemberg, Germany (MWK, grant 42-04HV.MED[16]/16/1). D.S.B. and R.B.F. would like to acknowledge support from the John D. and Catherine T. MacArthur Foundation, the Alfred P. Sloan Foundation, the Army Research Laboratory and the Army Research Office through contract numbers W911NF-10-2-0022 and W911NF-14-1-0679, the National Institute of Health (2-R01-DC-009209-11, 1R01HD086888-01, R01-MH107235, R01-MH107703, and R21-M MH-106799), the Office of Naval Research, and the National Science Foundation (BCS-1441502, CAREER PHY-1554488, and BCS-1631550). E.S. gratefully acknowledges grant support by the DFG (SCHW 1768/1-1) and BMBF (grant 01KU1905A). X.L.Z. is a Ph.D. scholarship awardee of the Chinese Scholarship Council. D.D. acknowledges grant support by the DFG (Du 354/10-1). G.P. has received funding from the European Union's Horizon 2020 research and innovation program under the Marie Skłodowska-Curie No. 798181: "IdentiFication of brain deveLopmental gene co-expression netwOrks to Understand RIsk for SchizopHrenia" (FLOURISH). The content of this paper is solely the responsibility of the authors and does not necessarily represent the official views of any of the funding agencies

## Author contributions

U.B. conceived and designed the study, performed experiments, analyzed data, provided code, wrote the manuscript. A.H. performed experiment, analyzed data, curated data, edited the manuscript. G.P. provided code, edited the manuscript. T.M. conceived the study, provided code, edited the manuscript. A.S. conceived the study, edited the manuscript. R.B. provided code, edited the manuscript, supervision. Z.Z. analyzed data, curated data, performed experiments. J.I.S. curated data, performed experiments, edited the manuscript. X.Z. curated data, analyzed data. K.S. performed experiments. J.C. curated data, analyzed data, edited the manuscript. G.B. provided resources, supervision, edited the manuscript. A.B. provided resources, supervision, edited the manuscript. D.D. supervision, edited the manuscript. F.P. conceived the study, provided code, supervision, edited the manuscript. E.S. curated data, provided resources, supervision, edited the manuscript. A.M.-L. conceived and designed study, provided resources, edited the manuscript. D.S.B. conceived and designed the study, provided resources, supervision, wrote the manuscript. H.T. conceived and designed the study, provided resources, supervision, wrote the manuscript.

## Funding

## Competing interests

A.M.-L. has received consultant fees from Blueprint Partnership, Boehringer Ingelheim, Daimler und Benz Stiftung, Elsevier, F. Hoffmann-La Roche, ICARE Schizophrenia, K. G. Jebsen Foundation, L.E.K Consulting, Lundbeck International Foundation (LINF), R. Adamczak, Roche Pharma, Science Foundation, Synapsis Foundation – Alzheimer Research Switzerland, System Analytics, and has received lectures including travel fees from Boehringer Ingelheim, Fama Public Relations, Institut d'investigacions Biomèdiques August Pi i Sunyer (IDIBAPS), Janssen-Cilag, Klinikum Christophsbad, Göppingen, Lilly Deutschland, Luzerner Psychiatrie, LVR Klinikum Düsseldorf, LWL PsychiatrieVerbund Westfalen-Lippe, Otsuka Pharmaceuticals, Reunions i Ciencia S. L., Spanish Society of Psychiatry, Südwestrundfunk Fernsehen, Stern TV, and Vitos Klinikum Kurhessen. A.B. has received consultant fees from Biogen and speaker fees from Lundbeck, Otsuka, Recordati, and Angelini. The remaining authors declare no competing interest.
