## [Peer Review File · Nature Communications]

Reviewers' comments:

Reviewer #1 (Remarks to the Author):

This paper by Braun et al describes an ambitious attempt to understand working memory dynamics, by employing network control theory. Specifically, they aim to investigate working memory as a switch between stable and transitional states mediated by dopaminergic reception in line with dual-state theory of prefrontal dopamine function. They model brain network dynamics during a simple motor task and a more cognitively demanding 2-back task, as a function of white matter connectivity tracts and regional control energy. They find that the 2-back task state is less stable than the control state and that the stability is associated with WM accuracy. Moreover, transitioning to the more demanding phase requires more control energy than the opposite. To further support the validity of their model, they use pharmacological, GWAS as well as patient studies. The PCI results relate stable states to the D1 pathway and transitions to the D2 pathway, the latter also backed by the pharmacological results with amisulpiride. Schizophrenia is used as a model of a perturbed system leading to decreased stability of working memory representations and a more diverse energy landscape.

While the questions posed in this paper are very timely and highly interesting, and the results mostly impressive, we have some difficulty seeing through the 'magic'.

From the main text, it is unclear how the key network control measures were derived, what was the rationale for their specific use, and for using (or methods of) DTI. When considering the supp mats, various key details become clear but many other details do not.

One specific concern is that there was no attempt to clarify the added value of assessing these indices over and above more standard overall BOLD signal patterns. The supp mats clarify that 'control energy can be interpreted as the effort of a brain region needed to steer the activity pattern of itself and its connected brain regions into the desired final activation state' but there is a need to clarify why shouldn't such effort not also translate into changes in standard BOLD signal.

Related to this, there was no theoretical attempt to conceptually relate the findings in terms of e.g. stability and required control energy to extensive prior evidence for increases in frontoparietal BOLD signal during cognitive effort and in default mode network regions during cognitive leisure, as well as greater frontoparietal BOLD signal when switching to an easy task vs switching to a difficult task (consider prior work in relation to the task-set inertia construct). This makes it hard to evaluate novel contribution of the findings e.g. in Figure 1.

While we can infer from the supp mats that network control analyses controlled for overall BOLD signal differences between task conditions, I am unclear whether this was done in a region- or voxel-specific manner or just by taking the average signal differences across the brain as a whole. The latter seems suboptimal.

We also felt the paper was poorly embedded in extant literature. No reference to extensive literature (empirical evidence from pharmacological work with nonhuman primates, rodents, humans as well as neural network modelling of the basal ganglia) on subcortical, striatal contributions to working memory gating and the flexibility/stability tradeoff (including a variety of studies reporting effects of D2 receptor agents on task switching).

Is it the case that effects are restricted to cortex?

Much of the key conclusion by the authors rely on indirect inferences about the role of prefrontal D1 and D2 receptors. However, without studying the supplementary materials, it is also unclear how the authors arrived at the genetic score, for example whether genetic data were actually obtained from the subjects whose n-back-related imaging data were considered. When considering the supp mats, it becomes clear that this analysis is based on GWAS data from only 63 subjects who also provided DTI and MRI data, but the paragraphs describing the procedures and rationales for achieving the 'PCI's' 'polygenic co-expression index calculation' are unclear. There is extensive reference to various other studies addressing effects of these PCIs on nback related BOLD signals, but this then raises the question again how do we evaluate the added and unique value of the control measure effects over and above voxel-specific BOLD measures.

Various of the effects (in the pharmacological and patient study) are borderline, hovering around the 0.05 threshold. I see no evidence of correction for multiple comparisons. This is an issue because emphasis is put on effects that are just below the 0.05 threshold (e.g. sulpiride) while some effects just above the 0.05 threshold hide in the supp mats (e.g. risperidone).

The title states 'Brain state stability during working memory is explained by network control theory'. However, for such a conclusion to carry weight, one would want to see that data are better fit from this theory than other theories. Comparison with alternative model(s) would be good.

The fact that the pharmacological study in healthy volunteers involved both administration of the selective agent amisulpiride as well as the more unselective (also serotonergic) agent risperidone was not mentioned in the main text, nor is the rationale for this design.

Also regarding the drug study we have access to actual datapoints reflecting only one fairly marginal effect on 'necessary control energy', but not to overall BOLD signals, other network control analysis outputs, like stability. This prevents the evaluation of the specificity of the effects. The same general point holds for the schizophrenia study.

In the patient study, medication status and disease severity are important potential confounds, and should be added as covariates to the models.

Overall, we think the authors packed take-home conclusions/results from too many experiments into one paper, preventing readers from evaluating the scientific basis of the conclusions. Using this format,

there was insufficient space for highlighting specific questions, predictions and rationales for each of the subcomponent elements of the studies. We strongly recommend that the paper is revised for a much more extensive format. The experiments are worth it.

Other minor comments:

Also, based on what reasoning were the first 5 genetic PCA components included as covariates of no interest? The supp matts refer to 'the PCA on the linkage-disequilibrium pruned set of autosomal SNPs'. Which PCA?

The 2-back is not process-pure and requires both flexibility and stability. This observation has motivated various researchers to decompose the task into its subcomponent elements, e.g. in the reference back paradigm, but there are other examples. While we recognize that the present analyses leveraged stationary patterns of activity to infer 'meta-level' brain states, this process impurity should be recognized explicitly.

Check typos in lines: 134, 142 and 150

Reviewer #2 (Remarks to the Author):

The manuscript "Brain state stability during working memory is explained by network control theory, modulated by dopamine D1/D2 receptor function, and diminished in schizophrenia" by Braun and colleagues presents a set of highly interesting and relevant investigations into the brain dynamics underlying working memory and their relationship to dopamine functioning, as indexed by whole-genome based co-expression indices of D1 and D2 receptors and pharmacological manipulation with amisulpride. A sample of schizophrenia patients was also included, in order to validate the results but also to demonstrate clinical implications of the findings.

The authors draw upon a very impressive array of different methods, including fMRI, connectomics, network control theory and brain network dynamics, molecular genetics and bioinformatics, pharmacological challenges, as well as clinical neuroscience. All methods are combined to assess theoretical predictions from the literature.

While the broad methodological approach is a clear strength of the work, it also comes with its own downsides: Large parts of the methodology are only superficially described, not well justified, and difficult to evaluate, even for a person who is familiar with almost all of the mentioned analyses.

As a general remark: I suggest to expand (at least) the supplement with much more information on the theoretical background, parameter choices, software code used, and additional explanations on the different procedures (connectome reconstruction, network control theory, genetic co-expression

analysis). I would further suggest to discuss results more in depth and with regard to methodological choices. The samples, recruitment, and experiments are described in great detail which I highly appreciate. But the manuscript would clearly benefit from including as much detail for all other analysis procedures as well. On the sample level, for instance, even miniscule detail on psychological control variables is given. Why do the other parts of the paper fall short of a similar level of detail?

General:

- The concepts of "energy", "energy barrier", and "energy landscape" require more elaboration. Particularly, since the energy concept is "loosely defined" (unquote) based on control signals that are themselves not introduced in depth
- Figure 1D: I find this analysis interesting but it is difficult to assess results in depth. In the main text it is mentioned that "other analyses" suggest fronto-parietal involvement in transitions, however, not much more information is given. A 20%-thresholding was applied to the figure. But information on this thresholding procedure, let alone a justification of this threshold is lacking.
- why is dopamine receptor expression a good candidate? It would be important to read more about the theoretical underpinnings and implications of the choice to study these particular variables.
- connectome reconstruction: crucial information on stopping rules / abortion criteria are missing (other than streamlines of <10 mm length)
- connectome reconstruction: Did you apply any thresholding to the FA matrices? Any particular reason to use FA and no alternative edge weight such as streamline volume density? [edit: Now that I read paragraph 9.4 this becomes a bit clearer].
- It is unclear whether participant level or group level connectome matrices were used. Either way: There is almost no information on thresholding, differences in network density, and information on how individual matrices were combined at the group level (if at all). I strongly advise the authors to be more specific here.
- network control theory: As with any new development in our fastly developing field I would appreciate more information here, rather than referring to previous work and stating the formulas. While it is quite straightforward to understand the conceptual logic behind the steps, a few more explanations and justifications throughout the section would make the whole manuscript more assessible for more readers.
- network control theory: More information on code availability and /or reference to the matlab functions
- genotyping: Step 1 and 2 of QC pipeline. What is meant by "numbers of SNPs"?
- Paragraph 8 statistical inference: It is somewhat confusing how "energy" and "activity" are conceptually intertwined. Would you mind clarifying?
- null models: Is there any reason the signed-network script was used? I am not sure how edge weights in FA weighted matrices could be negative.
- null models: Is there a way to quantify the interaction effect? Would be interesting to see how SCZ patients are different.
- null models: Please add information on the spatial randomization of brain activity.

Reviewer #3 (Remarks to the Author):

Thank you for inviting me to review this manuscript by Braun and colleagues, in which the authors apply a linear controllability analysis to structural brain imaging data from a large-sample of healthy individuals, and then use this information to infer the control theoretic principles required to shift between 0-back and 2-back activity on a cognitive task. As expected, shifting the brain into a cognitively-demanding 2-back task was associated with the need for greater control energy, suggesting that the state was more difficult to instantiate and maintain. The authors frame their results in the context of a previous hypothesis linking different cortical dopaminergic receptor families (D1 and D2 receptors) to deepening and flattening of an energy landscape, respectively. They test these predictions using a separate small cohort (N=16) of individuals that performed an N-back task following the administration of amisulpride (a D2-receptor agonist) and found results commensurate with their predictions. They also analysed a small (N=24) cohort of individual with clinically-diagnosed schizophrenia, and found similar impairments in control energy for the 2-back (but in this case, not the 0-back) condition

Overall, the manuscript was clearly written and technically impressive. There was a wide range of analytic techniques used, and as mentioned, analyses were conducted across multiple unique datasets. That said, I do have some reservations with the analyses as they are currently presented. I have outlined these concerns below, which I hope will help to improve the manuscript:

* The link between dopaminergic function and working memory is well-defined in prior literature, however I find the authors interpretation of the relationship between dopamine and the energy landscape somewhat puzzling. After reading the source material (Durstewitz and Seamans), I am confident that this issue relates to a focus on the impact of dopamine receptors on the activity patterns within the cerebral cortex. Although the impact of dopamine on the cortex is well-described using the energy landscape framework, I am less convinced that the same could be said for the effects of dopamine on the basal ganglia (which is their main site of action in the CNS, I might add). The circuitry of the basal ganglia and thalamus are quite distinct from the cerebral cortex, and there is good reason to believe that the effects of dopamine might be contrary to those acting locally on pyramidal cells in the cortex. For these reasons, I recommend that the authors make explicit their focus on the impact of dopamine on cortical circuits, and ensure that they are not incorrectly concluding that their effects should scale to the level of the entire system.

* I'm not sure that the term 'brain state' should be used to describe the spatial pattern of beta estimates from a general linear model (p5/6 Methods). These patterns instead represent a statistical map, which may or may not have recurred en masse, as one might expect for a 'brain state' -- indeed, it is quite possible for a region to be associated with a strong beta value from a GLM while only arising on a proportion of trials within an experiment. The authors may wish to choose a different term, or perhaps to confirm that the brain state was indeed a brain-wide state, and not just a statistical artefact.

* The authors mention that some of the subjects in the pharmacological study were administered Risperidone. Why were these data not reported?

* Is the choice to define the input matrix as an identity matrix well founded? This contrasts with my expectation that associative cortical regions should have a greater ability to influence sensory and motor regions of the cortex than vice versa. Would taking this into account change your results?

Minor:

* I couldn't find the value ' α ' defined on p7 of the Methods.

* I did not see the number of individuals with Schizophrenia mentioned in the main text.

Reviewers' comments:

Reviewer #1 (Remarks to the Author):

This paper by Braun et al describes an ambitious attempt to understand working memory dynamics, by employing network control theory. Specifically, they aim to investigate working memory as a switch between stable and transitional states mediated by dopaminergic reception in line with dual-state theory of prefrontal dopamine function. They model brain network dynamics during a simple motor task and a more cognitively demanding 2-back task, as a function of white matter connectivity tracts and regional control energy. They find that the 2-back task state is less stable than the control state and that the stability is associated with WM accuracy. Moreover, transitioning to the more demanding phase requires more control energy than the opposite. To further support the validity of their model, they use pharmacological, GWAS as well as patient studies. The PCI results relate stable states to the D1 pathway and transitions to the D2 pathway, the latter also backed by the pharmacological results with amisulpiride. Schizophrenia is used as a model of a perturbed system leading to decreased stability of working memory representations and a more diverse energy landscape.

While the questions posed in this paper are very timely and highly interesting, and the results mostly impressive, we have some difficulty seeing through the 'magic'.

1. From the main text, it is unclear how the key network control measures were derived, what was the rationale for their specific use, and for using (or methods of) DTI. When considering the supp mats, various key details become clear but many other details do not.

We thank the reviewers for this comment. The manuscript was originally submitted as a short communication format including strict word limitations, but the current requirements of Nature Communication allows for a more detailed and thorough description of the methods and ideas. Therefore, we have substantially extended and reorganized the manuscript to give the reader a more comprehensive but also more intuitive understanding of the key concepts and methods, in particular explaining the rationale for including DTI methods.

Specifically, we have added a figure explaining the key methods (Figure 1), and we have also added the following paragraphs to main manuscript:

page 3:

“The complex dynamics of state transitions unfold upon the underlying structural scaffold whose architecture shapes the structure-function relationship (17, 18) and constrains the dynamic repertoire enabling executive functioning (19-21).”

page 5:

“We define individual brain states as spatial patterns of β estimates associated with activity across brain regions of interest during a working memory condition (2-back) and during an attention control condition requiring motor response (0- back). It is important to note that our definition of brain states relates to the statistical spatial pattern of β estimates from a general linear model and does not reflect neuronal activity occurring en masse as, for example, in neurophysiological animal experiments. To quantify the energy efforts associated with a specific transition from an initial state x_0 to a target state x_T , we approximate brain dynamics locally by a simple linear dynamical system, $\dot{x}(t) = Ax(t) + Bu(t)$, where $x(t)$ is the brain state of the system, A is a structural connectome inferred from DTI data, u is the control input, and B is a matrix describing which regions enact control or receive control input. After finding the optimal control input u that enables a transition, the control energy of each node is calculated as the squared integral over time of u ; intuitively, this quantity measures the control input that the node has to exhibit to facilitate the transitions from the initial state to the target state. Similarly, the stability of a brain state can be defined as the inverse control energy needed to maintain in a specific state. In this framework, control energy can be interpreted as the effort of a brain region needed to steer the activity pattern of itself and its connected brain regions into the desired final

activation state; relatedly, stability can be interpreted as the effort of a brain region needed to maintain a given activity pattern of itself and its connected brain regions. For a more detailed mathematical description of the network control framework, please see the Methods section and Supplemental Information.”

Page 18 & 19:

“To model the transition between 0-back and 2-back brain states, we used the framework of optimal control, following prior work (23, 24, 76) implemented in MATLAB. Based on individual brain states $X=[x_1, \dots, x_n]$ (in our case simplified to $n = 2$ states: 0-back and 2-back, see above) and a structural brain network A for each subject, we approximated the local brain dynamics by a linear continuous-time equation, $\dot{x}(t) = Ax(t) + Bu(t)$, to model the flow among task-related brain activity states. In the model, $x(t)$ is the state of the system at time t , A is the wiring diagram of the underlying network, B denotes an input matrix defining the control nodes, and $u(t)$ is the time-dependent control signal (23, 41). Note that while the initial state (x_0) and the target state (x_T) are empirically defined, any states of the system at other times are virtual intermediate steps in the trajectories of the state-space model. In that state-space, we aim to identify a trajectory between state x_0 and state x_T that is minimal in terms of the necessary control input signals as well as the distance of the trajectory. This choice is motivated by two complementary ideas: first, that the brain minimizes its energy expenditure to perform that transition, and second, that optimal transitions between states should be non-random walks in state space. These notions can be formalized by defining an optimization problem that minimizes a given cost function. We define this cost function by the weighted sum of the energy cost of the transition and the integrated squared distance between the transition states and the target state.

The problem of finding an optimal control energy u^* that induces a trajectory from an initial state x_0 to a target state x_T reduces to the problem of finding an optimal solution to the minimization problem of the corresponding Hamiltonian:

$$(2) \min[H(p, x, u, t) = x^T x + \tilde{n} u^T u + p^T (Ax + Bu)] \quad (24).$$

The parameter \tilde{n} in this equation allows to penalize the energy used by the optimal input in relation to the deviation from the optimal trajectory when solving for the optimal control. As we had no specific hypothesis that either of these elements of the cost function should prevail, we used the default of $\tilde{n} = 1$.

By setting the input matrix $B = I_{N \times N}$, the identity matrix, we allow all brain regions to be independent controllers (41, 76). This is motivated by our analysis question to use a system-level and data-driven approach to identify regions contributing most to the transitions in an unbiased way in line with previous work (77-79).”

In addition, we have added a section giving further details regarding parameter choices to the Supplement. See SI Section 4. Network Control Theory.

2. One specific concern is that there was no attempt to clarify the added value of assessing these indices over and above more standard overall BOLD signal patterns. The supp mats clarify that ‘control energy can be interpreted as the effort of a brain region needed to steer the activity pattern of itself and its connected brain regions into the desired final activation state’ but there is a need to clarify why shouldn’t such effort not also translate into changes in standard BOLD signal.

We thank the reviewers for raising this important question. Indeed, the association of control energy and BOLD activity is complex and the added value of using the control theory framework requires further elaboration.

First, in the control theory framework, the control energy (u) of a node and the state of that same node (x) are highly interrelated. For example, if we consider a simplified system consisting of only one node, then the control energy E necessary to change the state of that node from an initial state (x_0) to a target state (x_T) is basically a function of the squared difference in that node’s state [$E \sim (x_0 - x_T)^2$]. As our definition of brain states is based on β estimates that depend on BOLD activity, in such a simplified system control energy would not give any additional information other than

the usual contrast images $\beta_{2\text{back}} - \beta_{0\text{back}}$. However, if we consider a more complex system with more than one node, and where all nodes are connected via either direct or indirect links as summarized in the connectivity matrix A , then the control energy of a single node is not a simple function of the squared differences in its state but additionally accounts for the influence of other connected neighbors.

Second, as the reviewers remark below, our definition of brain states is built on stationary activity patterns that summarize “meta-level” information about different cognitive processes. Here, it is important to note that our analysis was not intended (and because of the design of our N-back task also not suited) to study the switching between BOLD activity patterns related to cognitive flexibility and stability. Instead, we investigate processes at a more abstract, system-level perspective that might not find a direct expression in BOLD signal changes.

To demonstrate the additional value of our analytical approach to more conventional GLM approaches in regards to the reported neurobiological associations and group differences, we performed several additional analyses, which we describe below.

1) Association of control energy with DRD1 and DRD2 expression:

We extracted the parameter estimates from the respective peak voxels reported in Fazio *et al.* (1) at [-29 53 24] for DRD1 associated brain activity and Selvaggi *et al.* (2) at [-33 49 13] for DRD2 associated brain activity and included them as covariates in our main analysis. Importantly, our results were unchanged.

Specifically, regarding the stability, for the DRD1 score \rightarrow 0-back relation we found $b = 0.184$, $p = 0.036$, and for the DRD1 score \rightarrow 2-back relation, we found $b = 0.242$, $p = 0.008$. In this analysis, age, sex, brain activity, first 5 genetic PCA components and brain activity at [-29 53 24] were included as covariates of non-interest. Regarding the control energy, for the DRD2 score \rightarrow 0- to 2-back relation we obtained $b = -0.075$, $p = 0.042$, and for the DRD2 score \rightarrow 2- to 0-back relation we obtained $b = -0.139$, $p = 0.062$. Like the previous analysis, here we included age, sex, stability, difference in brain activity, first 5 genetic PCA components and brain activity at [-33 49 13] as covariates of non-interest.

These results demonstrate that our control measures explain additional variance beyond pure brain activation and are potentially associated with differential neurobiological mechanisms.

2) D2 receptor manipulation using Amisulpride:

We performed a conventional SPM analysis for Placebo versus Amisulpride, using the 2-back>0back contrast and included the same covariates as in our control theory analysis. We could not detect any significant activation differences, either for the Placebo > Amisulpride nor the opposite contrast, even at a lenient threshold of $P < 0.001$ uncorrected. These results demonstrate that our control measures can identify meaningful intervention effects in the absence of pure activation differences.

3) Stability and Control Energy in Schizophrenia:

We performed a conventional SPM analysis for schizophrenia patients versus healthy controls, using the 2-back>0back contrast and included the same covariates as in our control theory analysis. At a lenient threshold of $P < 0.001$ uncorrected, we detected two significant clusters of voxels in our target regions, comprising dorsolateral prefrontal cortex, hippocampus, and parietal cortex (HC > SZ: right hippocampus at [42 -37 -13] and HC<SZ: right parietal cortex at [51 -43 17]). We extracted the peak voxel β estimates and used partial correlations to demonstrate that the control energy and stability values of our analysis are not associated with the more

conventional activity measures. In particular, the β estimates at the right hippocampus [42 -37 -13] were not significantly correlated with 0-back stability ($r_{\text{par}} = -0.005$, $p = 0.961$), 2-back stability ($r_{\text{par}} = 0.139$, $p = 0.167$), 0- to 2-back control energy ($r_{\text{par}} = -0.115$, $p = 0.253$) or 2- to 0-back control energy ($r_{\text{par}} = 0.150$, $p = 0.135$). Furthermore, the β estimates at [51 -43 17] were not significantly correlated with 0-back stability ($r_{\text{par}} = -0.079$, $p = 0.431$), 2-back stability ($r_{\text{par}} = -0.106$, $p = 0.293$), 0- to 2-back control energy ($r_{\text{par}} = -0.144$, $p = 0.151$), or 2- to 0-back control energy ($r_{\text{par}} = -0.100$, $p = 0.318$). These results again demonstrate the relative independence of our control measures from more conventional group activation differences.

In the revised manuscript, we have added these additional analyses to the Supplemental Material. Together, we believe that these additional results demonstrate the additional value of our analytical approach to more conventional GLM approaches.

3. Related to this, there was no theoretical attempt to conceptually relate the findings in terms of e.g. stability and required control energy to extensive prior evidence for increases in frontoparietal BOLD signal during cognitive effort and in default mode network regions during cognitive leisure, as well as greater frontoparietal BOLD signal when switching to an easy task vs switching to a difficult task (consider prior work in relation to the task-set inertia construct). This makes it hard to evaluate novel contribution of the findings e.g. in Figure 1.

We thank the reviewers for this comment and agree that a conceptual discussion of the relation between our findings and prior work in task switching is of interest to the readers. Accordingly, we have added the following paragraphs to pages 10-11 of the main manuscript:

“Further, switching to the cognitively more demanding and less stable state required more control energy than the inverse transition. The direction of this difference suggests that the cognitively more demanding state was more difficult to access, a notion that is in line with the idea of associated switch costs when turning to more difficult tasks (42). Importantly, these analyses were performed while controlling for the effects of mean brain activation; therefore, the results cannot be explained as mere epiphenomena of global or local brain activity levels as measured by fMRI, which have previously been shown to be task- and cognitive load-dependent (36, 43). When examining the regional contributions of brain areas to the control of state switching, we are able to differentiate between (i) a universal set of brain regions mainly located in prefrontal and parietal cortices supporting both transitions, and (ii) medial structures in default-mode related areas that showed particularly strong involvement in 0- to 2-back transitions. These results are in line with previous predictions from network control studies indicating that PFC areas are essential for controlling transitions into hard-to-reach states (22, 44), and further support the assumed role of frontal-parietal circuits in steering brain dynamics (45) and their prominent role in shifting tasks (9,46-48). The findings also emphasize the importance of the coordinated behavior of brain systems commonly displaying deactivations during demanding cognitive tasks (49).”

4. While we can infer from the sup mats that network control analyses controlled for overall BOLD signal differences between task conditions, I am unclear whether this was done in a region- or voxel-specific manner or just by taking the average signal differences across the brain as a whole. The latter seems suboptimal.

We thank the reviewers for the question. In our analyses, we controlled for brain-wide differences in β estimates as a reflection of associated BOLD signal changes. This is necessary, as our primary interest is in the global system's effort that is needed to change brain activity patterns independent of global changes in the BOLD signal. Correcting for differences of activity on the individual node level would be infeasible,

as this would remove precisely the effects that we are interested in studying, namely the change in regional signal amplitude.

5. We also felt the paper was poorly embedded in extant literature. No reference to extensive literature (empirical evidence from pharmacological work with nonhuman primates, rodents, humans as well as neural network modelling of the basal ganglia) on subcortical, striatal contributions to working memory gating and the flexibility/stability tradeoff (including a variety of studies reporting effects of D2 receptor agents on task switching).

We thank the reviewers for this suggestion. The main focus of our paper was two-fold: first, we aimed to study prefrontal contributions to stability and state switching during working memory involving brain-wide activity patterns and secondly, the modulation of these mechanisms by prefrontal dopamine receptor signaling. We recognize and highly appreciate evolving efforts to study the influence of striatal dopamine on gating mechanisms contributing to the stability/flexibility tradeoff during working memory and task switching. However, as the reviewers themselves point out later, the paradigm employed here is not designed to disentangle the differential contribution of striatal activity (or dopamine) on stability or task switching during working memory. For that reason, we have kept our focus on cortical contributions to the process, but now explicitly mention that said focus and discuss the emerging field of striatal contributions to task switching and working memory. In the revised manuscript, we have added the following passages:

Page 3:

“Recent accounts extend the idea of dopamine’s impact on working memory from a local prefrontal to a brain-wide network perspective (5-7), emphasizing the dual role of dopamine in regulating the complex interplay between striatal and prefrontal circuits critical for balancing the stability-flexibility tradeoff. Indeed, several lines of research support the notion that dopamine actions in frontal-parietal regions contribute to both maintaining cortical representations (8-10) and the flexible switching between different representations (2, 11, 12). Notably, a large body of evidence further demonstrates that the latter process additionally involves striatal-cortical interactions (6, 9), suggesting a gating function of the striatum for cortical memory representations (7). These accounts highlight the contribution of widespread neural circuits and their regulation by dopamine to working memory.”

Page 11 + 12:

“Secondly, in line with the prediction of the dual-state theory of network function, we show that the ability to control brain dynamics during working memory is differentially modulated by D1/D2 dopamine receptor functioning. D1-receptor signaling in frontal circuits has been previously shown to facilitate working memory by tuning signal-to-noise ratios in pyramidal neurons (1, 50), enabling stable network activation patterns that support maintenance of neural representations (4, 51). In contrast, D2-receptor activation in PFC can lead to decreased GABA and NMDA receptor-related currents, thereby counteracting D1-receptor activity and ultimately enabling higher flexibility and switching between cognitive representations (4). While these insights focus on cortical microcircuits and are derived from animal studies and theoretical modelling, our results complement previous work demonstrating that similar principles govern the modulatory actions of D1 (and D2 receptors) at the macroscopic level of brain-wide networks, particularly in frontal-parietal circuits (6).

It is important to note that our data provide association of whole brain processes with polygenic co-expression indices derived from prefrontal areas, and that previous studies have demonstrated differential expression patterns of dopamine receptors for prefrontal and striatal areas as well as differential (and even antagonist) behavior of dopamine receptor stimulation in striatal and prefrontal circuits (6, 52). Therefore, it seems plausible that our results mainly reflect dopamine actions as observed in PFC-related circuits, which are also the dominant control nodes in our model facilitating both state transitions. Both observations would support a model of frontal-parietal circuits serving as hub regions modulated by D2-receptor signaling, which controls and facilitates the flexible adaptation of brain-wide activity and connectivity patterns (14, 46, 47, 53, 54). While our model concentrates on PFC related dopamine action, it does not exclude the increasingly important concept that emphasizes

the additional role of striatal input and output gating as dopamine-related mechanisms contributing to a stability-flexibility tradeoff critical for cognitive control, task-switching, and working memory (7, 55, 56). However, studying the differential contributions of striatal and frontal dopamine signaling on working memory in future studies will require a finer grained task design to disentangle the several cognitive subprocesses that are currently mingled in the two conditions of our N-back task.

The idea of dopamine related frontal-parietal circuits as important regions for flexibly controlling the reconfiguration of brain-wide activity patterns is further supported by our pharmacological intervention, where we observe a specific increase of control energy for switching brain activation patterns after D2-blockade, but no effect on the stability of these patterns. Here, future work quantifying brain-wide D1- and D2-receptor levels *in vivo* in combination with pharmacological manipulations would provide valuable and strongly needed data to disentangle the specific contributions of the spatial distribution and different receptor subtypes to working memory processes.”

6. Is it the case that effects are restricted to cortex?

We appreciate the reviewer’s question. The effects are not restricted to the cortex since our work is based on a brain parcellation spanning both cortical and subcortical areas. Importantly, we also replicate our results using a different whole-brain parcellation scheme to show the robustness of our findings. We described the details of that parcellation in the supplement and have added a line explicitly mentioning its coverage in the main manuscript:

Page 4: “Building on a brain parcellation spanning both cortical and subcortical areas (see Methods), ...”

7. Much of the key conclusion by the authors rely on indirect inferences about the role of prefrontal D1 and D2 receptors. However, without studying the supplementary materials, it is also unclear how the authors arrived at the genetic score, for example whether genetic data were actually obtained from the subjects whose n-back-related imaging data were considered. When considering the supp mats, it becomes clear that this analysis is based on GWAS data from only 63 subjects who also provided DTI and MRI data, but the paragraphs describing the procedures and rationales for achieving the ‘PCI’s’ ‘polygenic co-expression index calculation’ are unclear. There is extensive reference to various other studies addressing effects of these PCIs on nback related BOLD signals, but this then raises the question again how do we evaluate the added and unique value of the control measure effects over and above voxel-specific BOLD measures.

We apologize for the unclear description of the polygenic co-expression scores used in our analyses. The scores in our analysis are computed by a weighted sum of predefined single nucleotide polymorphisms (SNP), similar to a polygenic risk score. The scores are computed based on genome-wide SNP data for each individual and then used as a proxy of D1 and D2 receptor expression in the DLPFC of that individual. The weights and the set of SNPs contributing to both scores have been derived from relating co-expression patterns of DRD1 and DRD2 mRNA from post mortem brains in the dorso-lateral prefrontal cortex (Braincloud dataset) to genetic polymorphisms, as described in detail in previous publications (1-3). As described in the Supplementary Material, the initial samples for defining and replicating the calculation of these scores were much larger than the n = 63 in our study. The subsample in our study was determined by the number of healthy subjects that had whole-genome genotype data available. In the revised manuscript, we have added a clearer description and detailed explanation of how these scores were constructed.

First, on page 7 of the main manuscript, we have added a more detailed description of how we constructed these scores.

“To estimate individual prefrontal D1 receptor expression in each participant, we utilized methods relating prefrontal cortex D1 and D2 receptor expression to genetic variation in their co-expression partner, thereby enabling us to predict individual dopamine receptor expression levels from genotype data across the whole genome (26, 27). Specifically, previous work using weighted Gene Co-expression Network Analysis (28) applied on the Braincloud dataset of post-mortem DLPFC gene expression (29) had identified 67 non-overlapping sets of genes based on their expression pattern. The co-expression gene sets including DRD1 and DRD2 were summarized into Polygenic Co-expression Indices (PCIs) based on weighted SNPs that predicted co-expression of these genes. Based on these weighted sums of SNPs, we calculated individual PCI scores related to D1 and D2 receptor expression for a subset of 63 individuals for which whole genome genotyping data were available (for more details see Methods and SI).”

In addition, as mentioned above, we demonstrated the added value of our analysis, by extracting the BOLD measures from the voxels reported in Fazio et al (1) and Selvaggi et al (2) and including them as covariates in our main analysis. This did not change our results significantly. Specifically, regarding the stability, for the DRD1 score → 0-back relation we found $b = 0.184$, $p = 0.036$, and for the DRD1 score → 2-back relation, we found $b = 0.242$, $p = 0.008$. In this analysis, age, sex, brain activity, first 5 genetic PCA components and brain activity at [-29 53 24] as peak voxel reported in Fazio et al were included as covariates of non-interest. Regarding the control energy, for the DRD2 score → 0- to 2-back relation we obtained $b = -0.075$, $p = 0.042$, and for the DRD2 score → 2- to 0-back relation we obtained $b = -0.139$, $p = 0.062$. Like the previous analysis, here we included age, sex, stability, difference in brain activity, first 5 genetic PCA components and brain activity at [-33 49 13] as peak voxel reported in Selvaggi et al as covariates of non-interest.

These results demonstrate that our control measures explain additional variance beyond pure brain activation and are potentially associated with differential neurobiological mechanisms.

We have added these results demonstrating the added values of our analysis to the Supplemental information, Section 6.8.

8. Various of the effects (in the pharmacological and patient study) are borderline, hovering around the 0.05 threshold. I see no evidence of correction for multiple comparisons. This is an issue because emphasis is put on effects that are just below the 0.05 threshold (e.g. sulpiride) while some effects just above the 0.05 threshold hide in the supp mats (e.g. risperidone).

We thank the reviewers for this suggestion. Based on the theoretical and experimental rationale of study, our main hypothesis in the pharmacological study was that D2 receptor manipulation impacts transition energy, which we tested using amisulpride, as it has the highest selectivity for the D2 receptor. Because the data was acquired as part of a larger study and also included the acquisition of risperidone in the same subjects, we felt that for reasons of transparency we should analyze the data for risperidone as well; our hypothesis was that we would detect similar but weaker effects when using a less selective D2 receptor agent. We now mention these results in the main text and refer the readers to the extensive set of *post-hoc* control and specificity analyses that we were asked to perform to prove that our results are robust to parameter choices and specific to the variables presented in the main text.

9. The title states ‘Brain state stability during working memory is explained by network control theory’. However, for such a conclusion to carry weight, one would want to see that data are better fit from this theory than other theories. Comparison with alternative model(s) would be good.

We agree that this statement is overly bold, as we do not test different models here. Therefore, we have changed the title to:

“Brain state stability during working memory as assessed by network control theory, is modulated by dopamine D1/D2 receptor function, and diminished in schizophrenia”

10. The fact that the pharmacological study in healthy volunteers involved both administration of the selective agent amisulpiride as well as the more unselective (also serotonergic) agent risperidone was not mentioned in the main text, nor is the rationale for this design.

The pharmacological data were acquired as part of larger study testing the differential effects of a broad range of psychotropic drugs, including for example ketamine and scopolamine. In each group of healthy volunteers, two active substances (here: amisulpiride and risperidone) were tested in a randomized, three-arm, controlled trial. As our main goal in the current manuscript was to selectively perturb dopamine D2-receptor activity, we focused our analysis on the effects of amisulpiride, as it has the highest selectivity for the D2 receptor and therefore should have the strongest and cleanest effect. However, for reasons of transparency we analyzed the data for risperidone as well, hypothesizing that we would detect similar but weaker effects when using a less selective D2 receptor agent. We included these additional data in the Supplementary Material due to the length limitations of our original submission. Given the current, more generous word limit, we are happy to include the results for risperidone in the main manuscript.

Page 8:

“Importantly, we were able to trend-wise replicate the results of the pharmacological intervention with risperidone, a substance showing lower D2-receptor selectivity (repeated measures ANOVA with drug and transition as within-subject factors, main effect of drug: $F(1,10) = 3.490$, $p = 0.091$; repeated measures ANOVA with drug and stability as within-subject factors, main effect of drug: $F(1,8) = 1.057$, $p = 0.334$, see Table S3).”

11. Also regarding the drug study we have access to actual datapoints reflecting only one fairly marginal effect on ‘necessary control energy’, but not to overall BOLD signals, other network control analysis outputs, like stability. This prevents the evaluation of the specificity of the effects. The same general point holds for the schizophrenia study.

We have added plots for control energy, stability, and overall BOLD signals to the Supplementary Materials, section 7.2. and 7.3. For our pharmacological study, as reported in the manuscript and the SI, we find a significant main effect of drug for control energy but not for stability. In addition, we do not find a significant main effect of drug for overall BOLD signal (repeated measures ANOVA with drug and brain activity as within-subject factors, main effect of drug: $F(1,12) = 1.106$, $p = 0.314$, sex, and drug order as covariates of non-interest).

For our patient study, as reported in the manuscript and the SI, we find a significant main effect of patient group on the control energy of the transition from 0-back to 2-back and the stability of the 2-back state. Regarding brain activity, we find a trend-wise significant effect of patient group on the 0-back activity ($F(1,100) = 3.235$, $p = 0.075$, age, sex as covariates of non-interest) and a significant effect of patient group on the 2-back activity ($F(1,100) = 4.860$, $p = 0.030$, same covariates of non-interest). As we use brain activity and difference in brain activity in our main analyses as covariates of non-interest, these results supports the specificity of our results.

12. In the patient study, medication status and disease severity are important potential confounds, and should be added as covariates to the models.

In Section 6.5. of the Supplementary Materials, we have performed control analyses to demonstrate that medication status and disease severity/duration are not correlated with our primary control parameters. However, we did not add those potential confounds as covariates in our ANOVA analysis because doing so would significantly reduce our ability to detect group differences, as these presence and absence of these confounds are plausibly tight to group status (healthy controls received no medication, etc.).

Supplement, page 13+14:

“In patients, the potential relationship between control energy and stability, antipsychotic drug dose (expressed in chlorpromazine equivalents (CPZE), n=20), and clinical parameters (illness duration, illness severity as indexed by global functioning (GAF) and Positive and Negative Symptom Scale (PANSS)) were explored using a Pearson correlation. Neither the control energy for the 0-back to 2-back transition nor the opposite transition or the stability of either state were significantly associated with CPZE (N = 20, 0- to 2-back: $r = 0.078$, $p = 0.767$; 2- to 0-back: $r = 0.320$, $p = 0.210$; 0- back stability: $r = 0.150$, $p = 0.564$; 2- back stability: $r = 0.096$, $p = 0.713$), with illness duration (N = 23, 0- to 2-back: $r = 0.017$, $p = 0.937$; 2- to 0-back: $r = -0.226$, $p = 0.299$; 0- back stability: $r = 0.110$, $p = 0.644$; 2- back stability: $r = 0.281$, $p = 0.230$), or with GAF (N = 24, 0- to 2-back: $r = -0.086$, $p = 0.690$; 2- to 0-back: $r = -0.254$, $p = 0.230$; 0- back stability: $r = -0.135$, $p = 0.570$; 2- back stability: $r = 0.066$, $p = 0.793$).”

13. Overall, we think the authors packed take-home conclusions/results from too many experiments into one paper, preventing readers from evaluating the scientific basis of the conclusions. Using this format, there was insufficient space for highlighting specific questions, predictions and rationales for each of the subcomponent elements of the studies. We strongly recommend that the paper is revised for a much more extensive format. The experiments are worth it.

We thank the reviewers for their appreciation of our analyses and are confident that our significantly reworked and extended manuscript can provide a more detailed, clearer, and thorough picture of the rationale, experiments, and results to support our conclusions.

Other minor comments:

1. Also, based on what reasoning were the first 5 genetic PCA components included as covariates of no interest? The supp matts refer to ‘the PCA on the linkage-disequilibrium pruned set of autosomal SNPs’. Which PCA?

We thank the reviewer for this comment . A principal component analysis is applied to the GWAS data set after the included SNPs have been pruned based on linkage disequilibrium. The resulting set of principal components is used to infer genetic ancestry. Genetic ancestry or genetic population stratification is a critical source of error and false positive associations in GWAS and PRS analysis (4-6). Therefore, we included principal components reflecting genetic ancestry as covariates of no interest in our statistical models to control for population stratification. To balance rigorous control of population structure with overfitting of our model we decided to include the first five principle components.

2. The 2-back is not process-pure and requires both flexibility and stability. This observation has motivated various researchers to decompose the task into its subcomponent elements, e.g. in the reference back paradigm, but there are other examples. While we recognize that the present analyses leveraged stationary patterns of activity to infer ‘meta-level’ brain states, this process impurity should be recognized explicitly.

We agree that the current task design imposes several limitations on our analyses. In the revised manuscript, we now discuss the process-impurity and brain state definition explicitly in our limitations section.

Page 13:

“Firstly, to relate brain dynamics to cognitive function, we focus on discrete “meta-level” brain states where each state is summarized by a single brain activation pattern rather than a linear combination of multiple brain activity patterns. These brain states do not purely reflect a single process but instead involve several cognitive subprocesses. Future studies could use specific paradigms to disentangle these subprocesses, in combination with more direct and time-resolved measures of brain activity such as MEG.”

3. Check typos in lines: 134, 142 and 150

Thank you for spotting these typographical errors. We have corrected them in the revised manuscript.

Reviewer #2 (Remarks to the Author):

The manuscript "Brain state stability during working memory is explained by network control theory, modulated by dopamine D1/D2 receptor function, and diminished in schizophrenia" by Braun and colleagues presents a set of highly interesting and relevant investigations into the brain dynamics underlying working memory and their relationship to dopamine functioning, as indexed by whole-genome based co-expression indices of D1 and D2 receptors and pharmacological manipulation with amisulpride. A sample of schizophrenia patients was also included, in order to validate the results but also to demonstrate clinical implications of the findings.

The authors draw upon a very impressive array of different methods, including fMRI, connectomics, network control theory and brain network dynamics, molecular genetics and bioinformatics, pharmacological challenges, as well as clinical neuroscience. All methods are combined to assess theoretical predictions from the literature.

While the broad methodological approach is a clear strength of the work, it also comes with its own downsides: Large parts of the methodology are only superficially described, not well justified, and difficult to evaluate, even for a person who is familiar with almost all of the mentioned analyses.

As a general remark: I suggest to expand (at least) the supplement with much more information on the theoretical background, parameter choices, software code used, and additional explanations on the different procedures (connectome reconstruction, network control theory, genetic co-expression analysis). I would further suggest to discuss results more in depth and with regard to methodological choices. The samples, recruitment, and experiments are described in great detail which I highly appreciate. But the manuscript would clearly benefit from including as much detail for all other analysis procedures as well. On the sample level, for instance, even miniscule detail on psychological control variables is given. Why do the other parts of the paper fall short of a similar level of detail?

We thank the reviewer for their appreciation of our manuscript. We have now substantially restructured and expanded the manuscript and supplementary materials to include more details on the theoretical background, the parameter choices, and the methodological choices used in the study. We have also added more explanations and justifications for these choices, especially in regard to the more novel framework of network control theory.

General:

1. The concepts of "energy", "energy barrier", and "energy landscape" require more elaboration. Particularly, since the energy concept is "loosely defined" (unquote) based on control signals that are themselves not introduced in depth

We thank the reviewer for this comment and are sorry for the confusion. Our manuscript focuses on control energy, which quantifies the energetic effort needed to enable state transitions; that is, the effort necessary to drive the system from one activity state to another. In this framework, control energy is explicitly defined and can be calculated as the squared integral over time of the control. Our initial wording of "loosely defined" was admittedly poorly chosen, but reflected the fact that control energy in our case cannot be directly related to the physical definition of the word energy. Control energy is given in arbitrary units, and not in Joules; a mapping from the arbitrary units to physical units would be needed for us to determine a direct metabolic cost.

Additionally, the concepts and language of energy landscape and energy barrier are based on previous work in *non-linear* systems. Our work is built on a framework using *linear* system approaches, which make the computations of control signals

tractable. A direct comparison of linear and non-linear approaches is not possible, but a local approximation of a non-linear system (as the brain certainly is) to a linear system can be accurate (7, 8). In that context, an energy barrier between two states in the multi-dimensional state-space should be reflected in less control energy efforts to stabilize that state, but this relation has not yet been formalized mathematically.

However, we see that this wording led to confusion and we now explicitly explain how control energy is defined and calculated in the main text and the supplement, thereby highlighting these points:

E.g., supplement, page 7:

“Control energy for each node k_i , $i = 1 \dots m$ (m = total number of brain nodes), was defined as

$$(3) \quad E_{k_i} = \int_{t=0}^T \|u^*_{k_i}(t)\|^2 dt,$$

i.e. the squared integral over time of energy input that the node has to exhibit to facilitate the transitions from the initial state to the target state (9-11). While the neurobiological foundations of control properties in the brain are not yet well understood, and our framework for defining control energy is based on a linear systems approach and cannot be directly related to the physical definition of the word energy (with joule as a derived unit), this definition can be interpreted as the effort of a brain region needed to steer the activity pattern of itself and its connected brain regions into the desired final activation state, for example by tuning their internal firing or activity patterns by recurrent inhibitory connections.”

In addition, we have adapted a clearer separation of the word energy in the context of our dynamical model. We now refer to the input variables of our dynamical model (formerly baseline energy) as brain activity, while maintaining the terminology of control energy for the output of the control analysis (see response to comment #10 below).

2. Figure 1D: I find this analysis interesting but it is difficult to assess results in depth. In the main text it is mentioned that "other analyses" suggest frontal-parietal involvement in transitions, however, not much more information is given. A 20%-thresholding was applied to the figure. But information on this thresholding procedure, let alone a justification of this threshold is lacking.

We thank the reviewer for this comment. Please note that this figure does not illustrate the outcome of any of our main, hypothesis-driven and whole-brain network-based analyses. The purpose of this figure is largely descriptive and exploratory in nature, since we wanted to give readers the opportunity to visually explore the rather unique or rather shared contributions of different brain areas to the examined whole-brain network effects. As suggested, we have described the procedure in more detail in the main text. Specifically, and following Ref. (11) to quantify the contribution of each node to a transition, we have computed the control impact of each node by iteratively removing one node from the control set and recomputing the necessary control energy. By dividing the recomputed control energy E_{N-1} by the control energy needed when enabling the full control set E_N , we get a relative quantitative indicator of how much a specific node makes that transition harder or easier. Figure 1D is a visualization of the control impact for both transitions (0- to 2-back and *vice versa*); although the calculations were performed for all regions, here in the visualization we only show the 20% nodes with the highest impact for both transitions separately. We now explicitly mention the chosen threshold and exploratory nature of the figure in

the figure legend (see legend to Fig. 2 in the main text). Moreover, since we agree that the 20% threshold for illustration purposes is arbitrary, we now also provide additional figures for different thresholds as well as raw values in the supplement (see Fig. S4 in the supplement).

Main manuscript, page 6:

“Exploratory visualization of the 20% of brain regions that exhibited the highest control impact in both transitions suggested that prefrontal and parietal cortices may steer both types of transitions, while default mode areas may be preferentially important for switching to the more cognitively demanding state (**Fig. 2d**; see SI for illustration of alternative thresholds).“

Figure legend Fig 1:

“For illustrative purposes, we projected the computed control impact of each brain region for the respective transitions on a 3D structural template, displaying regions with the 20% highest control impact for each transition (see SI for illustration of alternative thresholds).“

3. why is dopamine receptor expression a good candidate? It would be important to read more about the theoretical underpinnings and implications of the choice to study these particular variables.

We thank the reviewer for this question. Dopamine receptor expression is a good candidate since a large body of evidence links dopamine to working memory and prefrontal circuit function (12-16) and several common single nucleotide polymorphisms that influence cortical (and other regional) dopamine receptor expression have been found to influence working memory performance and prefrontal activity and connectivity (17-20). In addition, recent results suggest that dopamine receptor expression can predict prefrontal activity (1-3), thus strengthening the idea that genetically-encoded expression of dopamine receptor levels in PFC influence prefrontal function.

In the revised manuscript, we have extended our description of the rationale for studying dopamine receptor expression in the context of working memory and prefrontal cortex function. We also now note the limitations of studying receptor expression as an indirect measure of dopamine function in the absence of *in-vivo* data on, for example, dopamine binding capacity that might give a more accurate and complex picture of dopamine signaling. In particular, we have added the following passages to the revised manuscript:

Page 3:

“Recent accounts extend the idea of dopamine’s impact on working memory from a local prefrontal to a brain-wide network perspective (5-7), emphasizing the dual role of dopamine in regulating the complex interplay between striatal and prefrontal circuits critical for balancing the stability-flexibility tradeoff. Indeed, several lines of research support the notion that dopamine actions in frontal-parietal regions contribute to both maintaining cortical representations (8-10), and the flexible switching between different representations (2, 11, 12). Notably, a large body of evidence further demonstrates that the latter process additionally involves striatal-cortical interactions (6, 9), suggesting a gating function of the striatum for cortical memory representations (7). These accounts highlight the contribution of widespread neural circuits and their regulation by dopamine to working memory.”.

Page 11 + 12:

“Secondly, in line with the prediction of the dual-state theory of network function, we show that the ability to control brain dynamics during working memory is differentially modulated by D1/D2 dopamine receptor functioning. D1-receptor signaling in frontal circuits has been previously shown to facilitate working memory by tuning signal-to-noise ratios in pyramidal neurons (1, 50), enabling stable

network activation patterns that support maintenance of neural representations (4, 51). In contrast, D2-receptor activation in PFC can lead to decreased GABA and NMDA receptor-related currents, thereby counteracting D1-receptor activity and ultimately enabling higher flexibility and switching between cognitive representations (4). While these insights focus on cortical microcircuits and are derived from animal studies and theoretical modelling, our results complement previous work demonstrating that similar principles govern the modulatory actions of D1 (and D2 receptors) at the macroscopic level of brain-wide networks, particularly in frontal-parietal circuits (6).

It is important to note that our data provide association of whole brain processes with polygenic co-expression indices derived from prefrontal areas, and that previous studies have demonstrated differential expression patterns of dopamine receptors for prefrontal and striatal areas as well as differential (and even antagonist) behavior of dopamine receptor stimulation in striatal and prefrontal circuits (6, 52). Therefore, it seems plausible that our results mainly reflect dopamine actions as observed in PFC-related circuits, which are also the dominant control nodes in our model facilitating both state transitions. Both observations would support a model of frontal-parietal circuits serving as hub regions modulated by D2-receptor signaling, which controls and facilitates the flexible adaptation of brain-wide activity and connectivity patterns (14, 46, 47, 53, 54). While our model concentrates on PFC related dopamine action, it does not exclude the increasingly important concept that emphasizes the additional role of striatal input and output gating as dopamine-related mechanisms contributing to a stability-flexibility tradeoff critical for cognitive control, task-switching and working memory (7, 55, 56). However, studying the differential contributions of striatal and frontal dopamine signaling on working memory in future studies will require a finer grained task design to disentangle the several cognitive subprocesses that are currently mingled in the two conditions of our N-back task.

The idea of dopamine related frontal-parietal circuits as important regions for flexibly controlling the reconfiguration of brain-wide activity patterns is further supported by our pharmacological intervention, where we observe a specific increase of control energy for switching brain activation patterns after D2-blockade, but no effect on the stability of these patterns. Here, future work quantifying brain-wide D1- and D2-receptor levels in vivo in combination with pharmacological manipulations would provide valuable and strongly needed data to disentangle the specific contributions of the spatial distribution and different receptor subtypes to working memory processes.

4. connectome reconstruction: crucial information on stopping rules / abortion criteria are missing (other than streamlines of <10 mm length)
We apologize for not including this important information in the previous manuscript. We have used default settings recommended in the documentation of DSI studio. In particular, a maximum angle of 75 degrees was applied as a termination criterion in addition to the minimal streamline length of 10 mm. We used a step size of 1mm and did not apply smoothing. In the revised manuscript, we have added these and other relevant details to the methods section; see “Connectome construction”.
5. connectome reconstruction: Did you apply any thresholding to the FA matrices? Any particular reason to use FA and no alternative edge weight such as streamline volume density? [edit: Now that I read paragraph 9.4 this becomes a bit clearer].

We appreciate the opportunity to clarify. Indeed, we used fractional anisotropy to weight the elements of the matrix, and also to index the fiber threshold. The value of this threshold was determined automatically using Otsu’s threshold multiplied by a factor of 0.6, which is the recommended default setting of DSI studio (21). To ensure that our results were stable across different estimates of anatomical connectivity, we used both fractional anisotropy and the number of streamlines normalized by the respective size of the regions as edges in the construction of structural connectivity matrices (22-27). Notably, we replicated all main results across these two complimentary edge definitions, thus confirming that our findings are robust to the measure of structural connectivity. For more details on these analyses and associated results, please see Supplementary Table 1.

6. It is unclear whether participant level or group level connectome [sic] matrices were used. Either way: There is almost no information on thresholding, differences in network density, and information on how individual matrices were combined at the group level (if at all). I strongly advise the authors to be more specific here.

We apologize for our unclear description here. For all parts of the analysis, except for the pharmacological data, we used individual reconstructed connectomes as outlined in the Methods section. Because diffusion data was not acquired in the pharmacological study, we used an averaged connectome across all healthy subjects in the associated main analysis. The average connectome was computed by averaging the FA values of all links following previous work (28, 29). In the revised manuscript, we now describe this procedure explicitly on page 8 of the main text:

“Importantly, because no DTI data was acquired in this study, we used a link-wise averaged connectivity matrix from all healthy subjects in study 1, following previous work, following previous work (30, 31).”

In addition, we have added a supplemental table quantifying several core graph metrics describing the individual structural connectivity matrices used in our analyses (see Table S2).

7. network control theory: As with any new development in our fastly developing field I would appreciate more information here, rather than referring to previous work and stating the formulas. While it is quite straightforward to understand the conceptual logic behind the steps, a few more explanations and justifications throughout the section would make the whole manuscript more assessable for more readers.

We thank the reviewer for this comment and are happy to further clarify our methods, particularly when they utilize more novel approaches. We have substantially reworked the manuscript and have included many more details and thorough explanations. In particular, we sought to give the reader a more intuitive understanding of the measures and parameters of network control theory. For ease of review, we provide the most heavily reworked passages below.

page 5:

“We define individual brain states as spatial patterns of β estimates associated with activity across brain regions of interest during a working memory condition (2-back) and during an attention control condition requiring motor response (0-back). It is important to note that our definition of brain states relates to the statistical spatial pattern of β estimates from a general linear model and does not reflect neuronal activity occurring en masse as, for example, in neurophysiological animal experiments. To quantify the energy efforts associated with a specific transition from an initial state x_0 to a target state x_T , we approximate brain dynamics locally by a simple linear dynamical system, $\dot{x}(t) = Ax(t) + Bu(t)$, where $x(t)$ is the brain state of the system, A is a structural connectome inferred from DTI data, u is the control input, and B is a matrix describing which regions enact control or receive control input. After finding the optimal control input u that enables a transition, the control energy of each node is calculated as the squared integral over time of u ; intuitively, this quantity measures the control input that the node has to exhibit to facilitate the transitions from the initial state to the target state. Similarly, the stability of a brain state can be defined as the inverse control energy needed to maintain in a specific state. In this framework, control energy can be interpreted as the effort of a brain region needed to steer the activity pattern of itself and its connected brain regions into the desired final activation state; relatedly, stability can be interpreted as the effort of a brain region needed to maintain a given activity pattern of itself and its connected brain regions. For a more detailed mathematical description of the network control framework, please see the Methods section and Supplemental Information.”

Page 18:

“In that state-space, we aim to identify a trajectory between state x_0 and state x_T that is minimal in terms of the necessary control input signals as well as the distance of the trajectory. This choice is motivated by two complimentary ideas: first that the brain minimizes its energy expenditure to perform that transition and second, that optimal transitions between states should be non-random walk in state space. These notions can be formalized by defining an optimization problem that minimizes a given cost function. We define this cost function by the weighted sum of the energy cost of the transition and the integrated squared distance between the transition states and the target state.”

In addition, we have added a section on parameter choices and justifications on the network control theory part to section 4 of the Supplementary Material.

8. network control theory: More information on code availability and /or reference to the matlab functions

We have included a statement on code availability as well as a link, where the MATLAB functions can be found.

9. genotyping: Step 1 and 2 of QC pipeline. What is meant by "numbers of SNPs"?

We would like to thank the reviewer for raising this question. The filtering step is part of the quality control (QC) procedure we used on the genotyping data set prior to calculating the PCI scores. The applied QC procedure is not specific for the hypotheses tested in the current study, but part of a commonly used QC pipeline for genome-wide genotyping data (30). In that pipeline, filtering of data is performed at two different levels: SNPs as well as samples which do not pass quality control are excluded from any further analyses.

One step within this quality control procedure is examining heterozygosity on the X chromosome in male participants. Given the haploid nature of the male X chromosome, the occurrence of a high number of heterozygous SNPs on the X chromosome in males is not plausible and most likely due to genotyping errors or erroneous assignments of sex in the metadata. Therefore, SNPs and samples with high heterozygosity on the male X-chromosome were excluded during QC.

10. Paragraph 8 statistical inference: It is somewhat confusing how "energy" and "activity" are conceptually intertwined. Would you mind clarifying?

We apologize for our unclear description and appreciate the opportunity to clarify. In this paragraph we use the term energy in reference to control energy as defined in the Methods section on network control analysis. The activity we refer to is task-related activation: as mentioned earlier and now described in greater depth, brain states are defined as region-wise GLM parameters without any units. Because the output and input to the dynamical model should, in theory, have the same units, we refer to both as energy. However, we recognize that such terminology could make it difficult to relate these concepts to neurobiological quantities. In the revised manuscript, we have therefore clarified our presentation by referring to the input variables (formerly baseline energy) as brain activity, while maintaining the terminology of control energy for the output of the control analysis. We believe this presentation will be much more intuitive to readers.

11. null models: Is there any reason the signed-network script was used? I am not sure how edge weights in FA weighted matrices could be negative.

Thank you for pointing this out. We have rerun our analyses using the *randmio_und_connected* function from the Brain Connectivity Toolbox and have updated the respective sections in the supplement. Importantly, this change did not alter our main results that human brain structural networks seem to be optimized to control brain state transitions in healthy controls, but annihilated the difference between randomized networks in schizophrenia (see below), see Supplemental Section 6.1.

12. null models: Is there a way to quantify the interaction effect? Would be interesting to see how SCZ patients are different.

We thank the reviewer for pointing this out. We have now directly tested for a group interaction effect on both of our control analyses. We observed no significant group by randomization effect for the randomized structural networks. However, we did observe a significant effect for randomized brain states, which could potentially suggest a more randomized activity pattern during working memory in schizophrenia. We have updated the Supplementary Material accordingly, see Supplemental Sections 6.1. and 6.2.

13. null models: Please add information on the spatial randomization of brain activity.

We apologize for the incomplete information. In the revised manuscript, we now include a detailed description on the algorithmic implementation, and we have updated the information regarding the number of repetitions.

“Randomization was done using the *randperm* function in MATLAB for the paired vectorized brain activation patterns (related to 0- and 2-back) followed by a recalculation of the control energy. This procedure was repeated 100 times and the averaged brain-wide control energy over all 100 iterations for each subject was used in the subsequent analysis.”

Reviewer #3 (Remarks to the Author):

Thank you for inviting me to review this manuscript by Braun and colleagues, in which the authors apply a linear controllability analysis to structural brain imaging data from a large-sample of healthy individuals, and then use this information to infer the control theoretic principles required to shift between 0-back and 2-back activity on a cognitive task. As expected, shifting the brain into a cognitively-demanding 2-back task was associated with the need for greater control energy, suggesting that the state was more difficult to instantiate and maintain. The authors frame their results in the context of a previous hypothesis linking different cortical dopaminergic receptor families (D1 and D2 receptors) to deepening and flattening of an energy landscape, respectively. They test these predictions using a separate small cohort (N=16) of individuals that performed an N-back task following the administration of amisulpride (a D2-receptor agonist) and found results commensurate with their predictions. They also analysed a small (N=24) cohort of individual with clinically-diagnosed schizophrenia, and found similar impairments in control energy for the 2-back (but in this case, not the 0-back) condition

Overall, the manuscript was clearly written and technically impressive. There was a wide range of analytic techniques used, and as mentioned, analyses were conducted across multiple unique datasets. That said, I do have some reservations with the analyses as they are currently presented. I have outlined these concerns below, which I hope will help to improve the manuscript:

1. The link between dopaminergic function and working memory is well-defined in prior literature, however I find the authors interpretation of the relationship between dopamine and the energy landscape somewhat puzzling. After reading the source material (Durstewitz and Seamans), I am confident that this issue relates to a focus on the impact of dopamine receptors on the activity patterns within the cerebral cortex. Although the impact of dopamine on the cortex is well-described using the energy landscape framework, I am less convinced that the same could be said for the effects of dopamine on the basal ganglia (which is their main site of action in the CNS, I might add). The circuitry of the basal ganglia and thalamus are quite distinct from the cerebral cortex, and there is good reason to believe that the effects of dopamine might be contrary to those acting locally on pyramidal cells in the cortex. For these reasons, I recommend that the authors make explicit their focus on the impact of dopamine on cortical circuits, and ensure that they are not incorrectly concluding that their effects should scale to the level of the entire system.

We thank the reviewer for this comment. As you correctly pointed out, the main focus of our paper was on the impact of prefrontal dopamine actions. We recognize and highly appreciate the evolving field of studying the influence of striatal dopamine on gating mechanisms contributing to working memory and task switching, but the design of the paradigm employed here is not suited to meaningfully disentangle the differential contribution of striatal activity (or dopamine) on stability or task switching during working memory. For that reason, we have kept our focus on cortical contributions to the process, but now explicitly mention said focus and discuss the emerging field of striatal gating.

In response to the reviewer's comment, we have added the following lines:

Page 3:

"Recent accounts extend the idea of dopamine's impact on working memory from a local prefrontal to a brain-wide network perspective (5-7), emphasizing the dual role of dopamine in regulating the complex interplay between striatal and prefrontal circuits critical for balancing the stability-flexibility tradeoff. Indeed, several lines of research support the notion that dopamine actions in frontal-parietal regions contribute to both maintaining cortical representations (8-10), and the flexible switching between different representations (2, 11, 12). Notably, a large body of evidence further demonstrates that the latter process additionally involves striatal-cortical interactions (6, 9), suggesting a gating

function of the striatum for cortical memory representations (7). These accounts highlight the contribution of widespread neural circuits and their regulation by dopamine to working memory.”

Page 11 + 12:

“Secondly, in line with the prediction of the dual-state theory of network function, we show that the ability to control brain dynamics during working memory is differentially modulated by D1/D2 dopamine receptor functioning. D1-receptor signaling in frontal circuits has been previously shown to facilitate working memory by tuning signal-to-noise ratios in pyramidal neurons (1, 52), enabling stable network activation patterns that support maintenance of neural representations (4, 53). In contrast, D2-receptor activation in PFC can lead to decreased GABA and NMDA receptor-related currents, thereby counteracting D1-receptor activity and ultimately enabling higher flexibility and switching between cognitive representations (4). While these insights focus on cortical microcircuits and are derived from animal studies and theoretical modelling, our results complement previous work demonstrating that similar principles govern the modulatory actions of D1 (and D2 receptors) at the macroscopic level of brain-wide networks, particularly in frontal-parietal circuits (6).

It is important to note that our data provide association of whole brain processes with polygenic co-expression indices derived from prefrontal areas, and that previous studies have demonstrated differential expression patterns of dopamine receptors for prefrontal and striatal areas as well as differential (and even antagonist) behavior of dopamine receptor stimulation in striatal and prefrontal circuits (6, 54). Therefore, it seems plausible that our results mainly reflect dopamine actions as observed in PFC-related circuits, which are also the dominant control nodes in our model facilitating both state transitions. Both observations would support a model of frontal-parietal circuits serving as hub regions modulated by D2-receptor signaling, which controls and facilitates the flexible adaptation of brain-wide activity and connectivity patterns (14, 48, 49, 55, 56). While our model concentrates on PFC related dopamine action, it does not exclude the increasingly important concept that emphasizes the additional role of striatal input and output gating as dopamine-related mechanisms contributing to a stability-flexibility tradeoff critical for cognitive control, task-switching and working memory (7, 57, 58). However, studying the differential contributions of striatal and frontal dopamine signaling on working memory in future studies will require a finer grained task design to disentangle the several cognitive subprocesses that are currently mingled in the two conditions of our N-back task.

The idea of dopamine related frontal-parietal circuits as important regions for flexibly controlling the reconfiguration of brain-wide activity patterns is further supported by our pharmacological intervention, where we observe a specific increase of control energy for switching brain activation patterns after D2-blockade, but no effect on the stability of these patterns. Here, future work quantifying brain-wide D1- and D2-receptor levels in vivo in combination with pharmacological manipulations would provide valuable and strongly needed data to disentangle the specific contributions of the spatial distribution and different receptor subtypes to working memory processes.”

2. I'm not sure that the term 'brain state' should be used to describe the spatial pattern of beta estimates from a general linear model (p5/6 Methods). These patterns instead represent a statistical map, which may or may not have recurred en masse, as one might expect for a 'brain state' -- indeed, it is quite possible for a region to be associated with a strong beta value from a GLM while only arising on a proportion of trials within an experiment. The authors may wish to choose a different term, or perhaps to confirm that the brain state was indeed a brain-wide state, and not just a statistical artefact.

We thank the reviewer for raising this important point. Indeed, our definition of brain states does not correspond to the definition usually used for example in neurophysiological animal experiments, but rather reflects statistical associations of meta-level brain activity. However, our definition does indeed correspond to the use of the term in the control theory framework, which defines a state as single point in a multi-dimensional state space. To be consistent its use in network control theory and to better enable readers from both fields to easily translate concepts between fields, we have used the term “brain state” to describe these activity patterns. To sensitize our readers to the different definition of the term “brain state”, we have added an

explicit definition of the term as we use it in our manuscript as well as a clear delineation of the more common definition in neurobiology.

Page 5:

“It is important to note that our definition of brain states relates to the statistical spatial pattern of beta estimates from a general linear model and does not reflect neuronal activity occurring *en masse* as, for example, in neurophysiological animal experiments.”

We further discuss this important limitation and suggest further directions for a more translatable definition of brain states on page 13:

“Firstly, to relate brain dynamics to cognitive function, we focus on discrete “meta-level” brain states where each state is summarized by a single brain activation pattern rather than a linear combination of multiple brain activity patterns. These brain states do not purely reflect a single process but instead involve several cognitive subprocesses. Future studies could use specific paradigms to disentangle these subprocesses, in combination with more direct and time-resolved measures of brain activity such as MEG.”

3. The authors mention that some of the subjects in the pharmacological study were administered Risperidone. Why were these data not reported?

Thank you for this question. To clarify, the pharmacological data were acquired as part of larger study testing the differential effects of a broad range of psychotropic drugs, including for example ketamine and scopolamine. In each group of healthy volunteers, two active substances in a randomized, three-armed, placebo-controlled crossover trial (here: amisulpride, risperidone, placebo). As our main goal in the current manuscript was to selectively perturb dopamine D2-receptor activity, we focused our analysis on the effects of amisulpride, as it has the highest selectivity for the D2 receptor and therefore should have the strongest and cleanest effect. In comparison to amisulpride, risperidone is a more unselective agent also influencing serotonergic receptors. As previously described in the supplement, we analyzed the risperidone data as well, thereby expecting a diminished effect on dopamine-related control properties. We now mention these results in the main text and provide further detailed information in the supplement.

Page 8:

“Importantly, we were able to trend-wise replicate the results of the pharmacological intervention with risperidone, a substance showing lower D2-receptor selectivity (repeated measures ANOVA with drug and transition as within-subject factors, main effect of drug: $F(1,10) = 3.490$, $p = 0.091$; repeated measures ANOVA with drug and stability as within-subject factors, main effect of drug: $F(1,8) = 1.057$, $p = 0.334$, see Table S3).”

4. Is the choice to define the input matrix as an identity matrix well founded? This contrasts with my expectation that associative cortical regions should have a greater ability to influence sensory and motor regions of the cortex than *vice versa*. Would taking this into account change your results?

Thank you for raising this important question. The choice to use the identity matrix as the input matrix was explicitly motivated by taking an data-driven and system-level approach, allowing us identify regions contributing most to the transitions in an unbiased fashion (31-33), which was our primary question. Restricting the set of nodes that can exert control, for example to cortical regions or the frontal-parietal control network, would ask different questions such as “Can these set of nodes control transitions?” or “Does this set of nodes require less/more energy to facilitate

a transition". While the former question involves the difficulty of identifying and dealing with an unstable dynamical system, the latter question does require appropriate null models (i.e. a comparison to a set of brain regions that for example has the same number of nodes while still being biologically meaningful). For those reason, we chose to use the identity matrix. We now explain that rationale explicitly in the Methods sections.

Methods section (page 19)

"By setting the input matrix $B = I_{N \times N}$, the identity matrix, we allow all brain regions to be independent controllers (43, 78). This is motivated by our analysis question to use a system-level and data-driven approach to identify regions contributing most to the transitions in an unbiased way in line with previous work (79-81)."

Minor:

1. I couldn't find the value ' $\tilde{\alpha}$ ' defined on p7 of the Methods.

We thank the reviewer for pointing this out. The parameter $\tilde{\alpha}$ in this equation allows to penalize the energy used by the optimal input (the second term of the internal) more or less than the deviation from the optimal trajectory (the first term of the integral) when solving for the optimal control. As we had no specific hypothesis that either of these parts of the cost function should prevail, we set $\tilde{\alpha} = 1$. We have added this explanation to the manuscript, see page 18:

"The parameter $\tilde{\alpha}$ in this equation allows to penalize the energy used by the optimal input in relation to the deviation from the optimal trajectory when solving for the optimal control. As we had no specific hypothesis that either of these elements of the cost function should prevail, we used the default of $\tilde{\alpha} = 1$."

2. I did not see the number of individuals with Schizophrenia mentioned in the main text.

Thank you. We apologize for not mentioning that information and have added the number of schizophrenia patients (n = 24) to the main text.

1. Fazio L, Pergola G, Papalino M, Di Carlo P, Monda A, Gelao B, et al. (2018): Transcriptomic context of DRD1 is associated with prefrontal activity and behavior during working memory. *Proc Natl Acad Sci U S A*. 115:5582-5587.
2. Selvaggi P, Pergola G, Gelao B, Di Carlo P, Nettis MA, Amico G, et al. (2019): Genetic Variation of a DRD2 Co-expression Network is Associated with Changes in Prefrontal Function After D2 Receptors Stimulation. *Cereb Cortex*. 29:1162-1173.
3. Pergola G, Di Carlo P, D'Ambrosio E, Gelao B, Fazio L, Papalino M, et al. (2017): DRD2 co-expression network and a related polygenic index predict imaging, behavioral and clinical phenotypes linked to schizophrenia. *Transl Psychiatry*. 7:e1006.
4. Price AL, Patterson NJ, Plenge RM, Weinblatt ME, Shadick NA, Reich D (2006): Principal components analysis corrects for stratification in genome-wide association studies. *Nat Genet*. 38:904-909.
5. Astle W, Balding DJ (2009): Population Structure and Cryptic Relatedness in Genetic Association Studies. *Statist Sci*. 24:451-471.
6. Martin AR, Gignoux CR, Walters RK, Wojcik GL, Neale BM, Gravel S, et al. (2017): Human Demographic History Impacts Genetic Risk Prediction across Diverse Populations. *The American Journal of Human Genetics*. 100:635-649.

7. Honey C, Sporns O, Cammoun L, Gigandet X, Thiran J, Meuli R, et al. (2009): Predicting human resting-state functional connectivity from structural connectivity. *Proc Natl Acad Sci U S A*. 106:2035-2040.
8. Galán RF (2008): On how network architecture determines the dominant patterns of spontaneous neural activity. *PLoS one*. 3.
9. Betzel RF, Gu S, Medaglia JD, Pasqualetti F, Bassett DS (2016): Optimally controlling the human connectome: the role of network topology. *Sci Rep*. 6:30770.
10. Cornblath EJ, Ashourvan A, Kim JZ, Betzel RF, Ciric R, Baum GL, et al. (2018): Context-dependent architecture of brain state dynamics is explained by white matter connectivity and theories of network control. *arXiv preprint arXiv:180902849*.
11. Gu S, Betzel RF, Mattar MG, Cieslak M, Delio PR, Grafton ST, et al. (2017): Optimal trajectories of brain state transitions. *Neuroimage*. 148:305-317.
12. Goldman-Rakic PS (1995): Cellular basis of working memory. *Neuron*. 14:477-485.
13. Williams GV, Goldman-Rakic PS (1995): Modulation of memory fields by dopamine D1 receptors in prefrontal cortex. *Nature*. 376:572-575.
14. Cools R (2019): Chemistry of the Adaptive Mind: Lessons from Dopamine. *Neuron*. 104:113-131.
15. Ott T, Nieder A (2019): Dopamine and Cognitive Control in Prefrontal Cortex. *Trends Cogn Sci*. 23:213-234.
16. Tost H, Braus D, Hakimi S, Ruf M, Vollmert C, Hohn F, et al. (2010): Acute D2 receptor blockade induces rapid, reversible remodeling in human cortical-striatal circuits. *Nat Neurosci*. 13:920-922.
17. Callicott JH, Mattay VS, Bertolino A, Finn K, Coppola R, Frank JA, et al. (1999): Physiological characteristics of capacity constraints in working memory as revealed by functional MRI. *Cerebral Cortex*. 9:20-26.
18. Zhang Y, Bertolino A, Fazio L, Blasi G, Rampino A, Romano R, et al. (2007): Polymorphisms in human dopamine D2 receptor gene affect gene expression, splicing, and neuronal activity during working memory. *Proc Natl Acad Sci U S A*. 104:20552-20557.
19. Meyer-Lindenberg A (2009): Neural connectivity as an intermediate phenotype: brain networks under genetic control. *Hum Brain Mapp*. 30:1938-1946.
20. Meyer-Lindenberg A (2010): From maps to mechanisms through neuroimaging of schizophrenia. *Nature*. 468:194-202.
21. Yeh FC, Verstynen TD, Wang Y, Fernandez-Miranda JC, Tseng WY (2013): Deterministic diffusion fiber tracking improved by quantitative anisotropy. *PLoS One*. 8:e80713.
22. Bohlken MM, Brouwer RM, Mandl RC, Van den Heuvel MP, Hedman AM, De Hert M, et al. (2016): Structural Brain Connectivity as a Genetic Marker for Schizophrenia. *JAMA psychiatry*. 73:11-19.
23. Masic B, Betzel RF, de Reus MA, van den Heuvel MP, Berman MG, McIntosh AR, et al. (2016): Network-Level Structure-Function Relationships in Human Neocortex. *Cereb Cortex*. 26:3285-3296.
24. van den Heuvel MP, de Reus MA, Feldman Barrett L, Scholtens LH, Coopmans FM, Schmidt R, et al. (2015): Comparison of diffusion tractography and tract-tracing measures of connectivity strength in rhesus macaque connectome. *Hum Brain Mapp*. 36:3064-3075.
25. van den Heuvel MP, Sporns O (2011): Rich-club organization of the human connectome. *J Neurosci*. 31:15775-15786.
26. Baker ST, Lubman DI, Yucel M, Allen NB, Whittle S, Fulcher BD, et al. (2015): Developmental Changes in Brain Network Hub Connectivity in Late Adolescence. *J Neurosci*. 35:9078-9087.

27. Li L, Rilling JK, Preuss TM, Glasser MF, Hu X (2012): The effects of connection reconstruction method on the interregional connectivity of brain networks via diffusion tractography. *Hum Brain Mapp.* 33:1894-1913.
28. Seguin C, van den Heuvel MP, Zalesky A (2018): Navigation of brain networks. *Proc Natl Acad Sci U S A.* 115:6297-6302.
29. Goñi J, van den Heuvel MP, Avena-Koenigsberger A, de Mendizabal NV, Betzel RF, Griffa A, et al. (2014): Resting-brain functional connectivity predicted by analytic measures of network communication. *Proceedings of the National Academy of Sciences.* 111:833-838.
30. Chen J, Lippold D, Frank J, Rayner W, Meyer-Lindenberg A, Schwarz E (2019): Gimpute: an efficient genetic data imputation pipeline. *Bioinformatics.* 35:1433-1435.
31. Cui Z, Stiso J, Baum GL, Kim JZ, Roalf DR, Betzel RF, et al. (2020): Optimization of energy state transition trajectory supports the development of executive function during youth. *Elife.* 9.
32. Karrer TM, Kim JZ, Stiso J, Kahn AE, Pasqualetti F, Habel U, et al. (2020): A practical guide to methodological considerations in the controllability of structural brain networks. *J Neural Eng.*
33. Stiso J, Khambhati AN, Menara T, Kahn AE, Stein JM, Das SR, et al. (2019): White Matter Network Architecture Guides Direct Electrical Stimulation through Optimal State Transitions. *Cell Rep.* 28:2554-2566 e2557.

REVIEWER COMMENTS

Reviewer #1 (Remarks to the Author):

Many thanks for this revision and rebuttal. The new manuscript does a much better job at detailing the methods (including how to arrive at control energy measures, polygenic coexpression scores etc), and its specific rationales. Its embedding within existing literature is also much clarified. The title now does justice to the content of the paper. I also appreciate the full reporting of the pharmacological study including the risperidone effects as well as the supplementary analyses showing the added value of these control energy measures over and above more standard task contrast-related BOLD signals.

I do think the manuscript itself would benefit from inclusion of something similar to the clarification provided in the rebuttal letter and here:

'First, in the control theory framework, the control energy (u) of a node and the state of that same node (x) are highly interrelated. For example, if we consider a simplified system consisting of only one node, then the control energy E necessary to change the state of that node from an initial state (x_0) to a target state (x_T) is basically a function of the squared difference in that node's state [$E \sim (x_0 - x_T)^2$]. As our definition of brain states is based on \hat{B} estimates that depend on BOLD activity, in such a simplified system control energy would not give any additional information other than the usual contrast images \hat{B}^2 - \hat{B}^0 . However, if we consider a more complex system with more than one node, and where all nodes are connected via either direct or indirect links as summarized in the connectivity matrix A , then the control energy of a single node is not a simple function of the squared differences in its state but additionally accounts for the influence of other connected neighbors.'

I also think that a cautionary note regarding the potential confounds of medication status and disease severity should be added given that lack of between-subject correlations in small samples provide weak proof of evidence for a null effect

Re my other comments:

Why was inclusion of the first 5 principle components considered to optimally balance rigorous control of population structure with overfitting of the model?

Reviewer #2 (Remarks to the Author):

I would like to start my review by expressing my appreciation to the authors' effort with this thorough revision. The manuscript has benefited tremendously from the additional detail, the intuitive explanations, and the provision of the matlab code. Most things are much clearer now and it is easier to follow all different analyses steps. This is impressive work!

The genetic, connectomic, and neuroimaging (including patients and pharmacological) analyses are clear

by now and I have only few further questions. My most central question is on the concepts and procedures of the main network control analysis.

The authors apply a new framework to analyze task fMRI data and - both importantly and impressively - show that their approach reveals incremental information that traditional mass-univariate analyses are not able to pick up. But I am wondering if it is really justified to make so many recurrences to network control theory. I find this difficult to explain, but please let me try: The data are a traditional blocked fMRI paradigm with two conditions, each associated with its spatial pattern of task-related brain responses, that the authors regard as brain states. While traditional analyses would now contrast the two conditions / state through a mass-univariate NHST to identify regional differences (i.e. which brain regions differ in task-evoked activity), the authors ask which brain regions contribute most in maintaining and switching states. To do this, they exploit information from the white matter scaffold (dti data, connectome). The main analysis is thus a linear mapping from state x_0 to x_t (which is the spatial distribution of task-evoked brain responses in the two conditions) that finds a vector u whose elements indicate for each brain region how much it contributes to the transition (or maintenance), given the constraints from the white matter scaffold. This problem is solved by a set of methods from network control theory: A hypothetical trajectory is assumed through which x_0 evolves into x_t . This involves a lot of assumptions, particularly to define (assumed) equivalences between the data and the constructs from control theory (state definition) and to make the problem computationally feasible (linear and continuous transitions). These assumptions are of course well explained and reasonable.

What I am struggling with is a clear distinction between the actual data level and the conceptual level from control theory. Sometimes I am under the impression that - just as I tried to summarize above - control theory is applied as a toolbox to solve a cognitive neuroscience neuroimaging problem. This is a great idea - we might need an assumption and maybe a simplification here and there, but it's stringent, reasonable, and informative.

Sometimes, however, I am also under the impression that control theory with its concepts of states and energy might be taken too much as given prior ground truth and the neuroimaging data are squeezed into its 'theoretical corset' with assumptions that then justify the application of the theory and validate the theory at the same time.

I would appreciate if the authors could elaborate along these lines. Maybe a bit more information (see also concrete questions below) would be helpful here. I believe that our field is in grave need of new methods to study the relationship between brain activity during tasks and brain connectivity, and I also believe that new conceptual frameworks and theories will be needed to advance cognitive neuroscience. I regard the present work as highly important and innovative contribution towards this goal. I would suggest to make sure that others will be able to use this work as reference for further investigations.

Detailed questions:

Line 490 (formula 5): Why is stability the *inverse* of the control energy required to maintain a state ($x_t = x_t$)?

Connectome reconstruction: I find 75 degrees quite lenient for a stopping criterion. Other work often uses a 45 degree criterion. Of course, it is difficult to validate parameter choices. I am just curious to learn the authors' motivation for this threshold?

Supplementary Material:

Paragraph 4.3. Stabilization of the dynamical system happens somewhat out of the blue. How is it that normalizing the system prevents implausible states? And does A denote the structural connectivity matrix? Is the stabilized or the unstabilized matrix passed into the `optim_fun.m` function?

4.4. Time horizon: $T = 1$: What unit is 1? I understand from the code, that different intermediate states are calculated between x_0 and x_t , from $T=0$ to $T=1$ in steps of .001. But this does not help understanding this parameter choice. Also: What is the rho parameter, and why is it set to 1? The input matrix S is likely to be `zeros(numNodes,numNodes)`?

In line 446 it reads that B denotes an input matrix defining the control nodes. Line 116 reads "B is a matrix describing which regions enact control or receive control input".

I find the definition in the methods section more appropriate: How can you distinguish between sending (control) and receiving nodes if not using some measure of effective connectivity?

other issues:

Is the following sentence correct? Are we talking about a diversified energy landscape or a less diversified energy landscape?

"In a diversified energy landscape, we expected that the variation of trajectories around the minimum-energy trajectory should be larger than expected in the less diversified energy landscape in schizophrenia, implying that small perturbations may have a more substantial impact.

Figure S4: Is there any reason that subcortical regions were omitted from the figure?

Reviewer #3 (Remarks to the Author):

The authors have adequately addressed my concerns.

REVIEWER COMMENTS

Reviewer #1 (Remarks to the Author):

Many thanks for this revision and rebuttal. The new manuscript does a much better job at detailing the methods (including how to arrive at control energy measures, polygenic coexpression scores etc), and its specific rationales. Its embedding within existing literature is also much clarified. The title now does justice to the content of the paper. I also appreciate the full reporting of the pharmacological study including the risperidone effects as well as the supplementary analyses showing the added value of these control energy measures over and above more standard task contrast-related BOLD signals.

We thank the reviewer for their appreciation of our revisions and additions to the manuscript.

1. I do think the manuscript itself would benefit from inclusion of something similar to the clarification provided in the rebuttal letter and here:
'First, in the control theory framework, the control energy (u) of a node and the state of that same node (x) are highly interrelated. For example, if we consider a simplified system consisting of only one node, then the control energy E necessary to change the state of that node from an initial state (x_0) to a target state (x_T) is basically a function of the squared difference in that node's state [$E \sim (x_0 - x_T)^2$]. As our definition of brain states is based on β estimates that depend on BOLD activity, in such a simplified system control energy would not give any additional information other than the usual contrast images $\beta_{2back} - \beta_{0back}$. However, if we consider a more complex system with more than one node, and where all nodes are connected via either direct or indirect links as summarized in the connectivity matrix A , then the control energy of a single node is not a simple function of the squared differences in its state but additionally accounts for the influence of other connected neighbors.'

We thank the reviewer for this suggestion. We have added these sentences to the respective section in the Supplemental Information, and we now refer to that section in the main text.

2. I also think that a cautionary note regarding the potential confounds of medication status and disease severity should be added given that lack of between-subject correlations in small samples provide weak proof of evidence for a null effect.

We have mentioned this as a limitation in the main text and now explicitly highlight the difficulty of between-subject correlations in small samples in the relevant section of the Supplemental Information.

Page 14:

"Thirdly, we cannot exclude the possibility that disorder severity, duration, symptoms, or medication may have influenced network dynamics in schizophrenia patients, ~~although our supplemental analyses do not support this conclusion~~ (see SI)."

SI page 8:

“Please note, that a lack of between-subject correlations in small samples can only provide weak proof of evidence for a null effect.”

3. Re my other comments:

Why was inclusion of the first 5 principle components considered to optimally balance rigorous control of population structure with overfitting of the model?

The exact number of principal components used varies between studies, ranging from none up to 10 (1, 2) with many imaging studies using an intermediate number of components (3, 4). Our choice of using the first five principal components was purely based on theoretical assumptions that an intermediate number of components should provide a decent balance between control for population structure and model overfitting, given the loss of degrees of freedom when adding more covariates of non-interest to our model given our sample size. Perhaps more importantly, however, our results are stable to variation in the number of included components, for example: DRD1 predicting 2-back stability (linear regression model, same covariates as in the main paper)

No PCA component as a covariate: $b = 0.226$, $p = 0.008$;

1 PCA component as a covariate: $b = 0.251$, $p = 0.004$;

2 PCA components as covariates: $b = 0.246$, $p = 0.005$;

3 PCA components as covariates: $b = 0.253$, $p = 0.005$;

4 PCA components as covariates: $b = 0.254$, $p = 0.005$;

5 PCA components as covariates: $b = 0.242$, $p = 0.007$.

Reviewer #2 (Remarks to the Author):

I would like to start my review by expressing my appreciation to the authors' effort with this thorough revision. The manuscript has benefited tremendously from the additional detail, the intuitive explanations, and the provision of the matlab code. Most things are much clearer now and it is easier to follow all different analyses steps. This is impressive work!

The genetic, connectomic, and neuroimaging (including patients and pharmacological) analyses are clear by now and I have only few further questions. My most central question is on the concepts and procedures of the main network control analysis.

We thank the reviewer for the generous commendation of our revised manuscript, and for their helpful feedback that served to strengthen our presentation. We are also very happy to address the reviewer's remaining concerns.

1. The authors apply a new framework to analyze task fMRI data and - both importantly and impressively - show that their approach reveals incremental information that traditional mass-univariate analyses are not able to pick up. But I am wondering if it is really justified to make so many recurrences to network control theory. I find this difficult to explain, but please let me try: The data are a traditional blocked fMRI paradigm with two conditions, each associated with its spatial pattern of task-related brain responses, that the authors regard as brain states. While traditional analyses would now contrast the two conditions / state through a mass-univariate NHST to identify regional differences (i.e. which brain regions differ in task-evoked activity), the authors ask which brain regions contribute most in maintaining and switching states. To do this, they exploit information from the white matter scaffold (dti data, connectome). The main analyses is thus a linear mapping from state x_0 to x_t (which is the spatial distribution of task-evoked brain responses in the two conditions) that finds a vector u whose elements indicate for each brain region how much it contributes to the transition (or maintenance), given the constraints from the white matter scaffold. This problem is solved by a set of methods from network control theory: A hypothetical trajectory is assumed through which x_0 evolves into x_t . This involves a lot of assumptions, particularly to define (assumed) equivalences between the data and the constructs from control theory (state definition) and to make the problem computationally feasible (linear and continuous transitions). These assumptions are of course well explained and reasonable.

What I am struggling with is a clear distinction between the actual data level and the conceptual level from control theory. Sometimes I am under the impression that - just as I tried to summarize above - control theory is applied as a toolbox to solve a cognitive neuroscience neuroimaging problem. This is a great idea - we might need an assumption and maybe a simplification here and there, but it's stringent, reasonable, and informative.

Sometimes, however, I am also under the impression that control theory with its concepts of states and energy might be taken too much as given prior ground truth and the neuroimaging data are squeezed into its 'theoretical corset' with assumptions that then justify the application of the theory and validate the theory at the same time.

I would appreciate if the authors could elaborate along these lines. Maybe a bit more information (see also concrete questions below) would be helpful here. I believe that our field is in grave need of new methods to study the relationship between brain activity during tasks and brain connectivity, and I also believe that new conceptual frameworks and theories will be needed to advance cognitive neuroscience. I regard the present work as highly important and innovative contribution towards this goal. I would suggest to make sure that others will be able to use this work as reference for further investigations.

We thank the author for this interesting and important comment. Indeed, in our analysis, we apply control theory as a statistical and theoretical tool to answer questions about neurobiological properties of brain function. The hypotheses that we aim to test are based on the dual-state theory framework, which also uses the terminology of brain states and energy. Translating and transferring across these three levels (control theory as a statistical tool, dual-state theory as a non-linear theoretical framework, brain imaging data defining meta-level brain states) is challenging and requires (reasonable) simplifications. We also appreciate the reviewer's aim to ensure that others will be able to use this work as a reference for further investigations. With this aim in mind, we have made several additions and changes to the main text that serve to underscore the appropriate distinctions between these three levels. We highlight the utility of network control theory as a *toolkit* to address many other sorts of questions in cognitive neuroscience in the future. In addition, we have (already previously) agreed to publish the rebuttal letter alongside the manuscript, so that interested readers can follow the discussion here.

Page 4:

*"A promising **framework tool** to study these questions **and concepts derived from the dual-state theory** is network control theory (NCT)."*

*"Within this **statistical** framework, we test the following predictions of the dual-state theory of brain network function and evaluate their implications for schizophrenia."*

Page 10:

*"We provide evidence that the stability of and switching between global brain activation patterns during working memory can be meaningfully **explained assessed** by network control theory"*

In addition, we have added a paragraph discussing those ideas in more depth to the supplemental information, **page SI 5 + 6:**

4.6. On the use of control theory as a statistical framework

In our analysis, we apply control theory as a statistical and theoretical tool to answer questions based on the theoretical "dual-state" framework regarding neurobiological properties of brain function. Translating and transferring across these three levels (control theory as a statistical tool, dual-state theory as a non-linear theoretical framework, brain imaging data defining meta-level brain states) is challenging and requires (reasonable) simplifications. The hypotheses that we aim to test are based on the dual-state theory framework, which also uses the

terminology of brain states and energy. In this framework, states and transitions are based on non-linear dynamics, corresponding to attractor basins, which translate to stable reoccurring activation patterns in neuronal ensembles (16-18). Abstracting these concepts to large-scale dynamics of brain macro-circuits provides the underlying basis for the idea that we aim to investigate here: relatively stable “meta”-level brain activation patterns as identified by neuroimaging (including all the caveats of the assumption of stationarity of brain activations measured by functional magnetic resonance imaging) populate a state-space for which we aim to identify the brain regions that are responsible for maintaining and shifting those activation patterns. To answer these cognitive neuroscience questions, we use network control theory as a toolkit that makes these questions computationally tractable in a linear dynamical system framework enabling us to quantify the associated “energy cost” of transitions on a brain region level. This effort requires certain (reasonable) assumptions, in particular to assume an equivalence between states defined by neuroimaging and states defined in the control theory framework, as well linear and continuous transitions between those states. Future work integrating biophysical models of task-induced brain activity in combination with network control theory and tailored imaging paradigms is critically needed to provide further evidence for the assumed relationships (and distinctions) between actual data, network control tools, and the theoretical framework.”

Detailed questions:

2. Line 490 (formula 5): Why is stability the *inverse* of the control energy required to maintain a state ($x_t = x_t$)?

We thank the reviewer for the question. The energy required to maintain a state is inversely related to the distance of the state from a local minimum on the energy landscape. States that are distant from a local minimum will also dissipate quickly under spontaneous activity, while states at the local minimum will not. Thus, the temporal stability of a state is inversely related to the control energy required to maintain a state (5). This inverse relationship is at once intuitive and relevant to the concepts of the dual-state theory.

We have expanded the respective section in the methods and also point the interested reader to more didactic resources that provide further and more detailed explanations:

Page 20:

“The rationale here is, that the energy required to maintain a state is inversely related to the distance of the state from a local minimum on the energy landscape. States that are distant from a local minimum will also dissipate quickly under spontaneous activity, while states at the local minimum will not. Thus, the temporal stability of a state is inversely related to the control energy required to maintain a state. For further details, we refer the interested reader to Refs. (23, 80, 82, 83).”

3. Connectome reconstruction: I find 75 degrees quite lenient for a stopping criterion. Other work often uses a 45 degree criterion. Of course, it is difficult to validate parameter choices. I am just curious to learn the authors' motivation for this threshold?

We agree that 75 is at the upper portion of the distribution of stopping criteria found in the literature, which ranges between 15 to 90 degrees. As the reviewer remarks correctly, it is difficult to find the optimal or “correct” parameter choice due to the lack of our knowledge of the ground truth. We therefore have chosen to use the default parameter recommend by DSI studio and then confirmed that the resulting brain networks looked anatomically plausible.

4. Supplementary Material:

Paragraph 4.3. Stabilization of the dynamical system happens somewhat out of the blue. How is it that normalizing the system prevents implausible states? And does A denote the structural connectivity matrix? Is the stabilized or the unstabilized matrix passed into the `optim_fun.m` function?

The section that the reviewer refers to was initially added to the manuscript as a guide for potential future user of network control theory. When we say that we stabilize the system, we mean that we decrease the average weight of the connectome to prevent brain states from diverging over time. If we did not stabilize the matrix, then brain states would be allowed to increase in their levels of activity uncontrollably. We know that such uncontrolled levels of activity are not neurobiologically plausible, due to clear metabolic, electrical, and other physical constraints. We wish to be clear, however, that by removing the possibility of these hyper activity states, we do not necessarily ensure that we have removed all (potentially other types of) implausible states. Further work integrating experiment and theory is needed to more clearly define types of implausible states, and their respective mechanisms (e.g., metabolic, electrical, informational, or other physical constraints).

To address the reviewer’s later questions, here A does indeed denote the structural connectivity matrix. The `optim_fun.m` function needs to be given the stabilized structural connectivity matrix in order to find an optimal open-loop control. We have added these clarifications to the respective paragraphs and to the README file of the MATLAB function, and have also provided references to several papers that provide a more in-depth introduction to network control theory and its use in the context of brain networks.

SI page 5:

“... A denotes the structural connectivity matrix...”

“[...] We therefore chose to normalize the system by decreasing the average weight of the connectome such that it goes to zero over time [...] Within the range of brain states that converge to zero over time, we cannot make statements regarding whether any of these intermediate brain states are biologically plausible or are realized in human brains. Further work integrating experiment and theory is needed to more clearly define types of implausible states, and their respective mechanisms (e.g., metabolic, electrical, informational,

or other physical constraints). A more in-depth and mathematical introduction and discussion can be found in Refs. (12-14).”

README file:

A (NxN) stable structural connectivity matrix, e.g. stabilized
 $A = (A / (|\lambda(A)_{\max}| + 1)) - 1$

5. 4.4. Time horizon: $T = 1$: What unit is 1? I understand from the code, that different intermediate states are calculated between x_0 and x_t , from $T=1$ to $T=1$ in steps of .001. But this does not help understanding this parameter choice. Also: What is the rho parameter, and why is it set to 1? The input matrix S is likely to be zeros(numNodes,numNodes)?

The parameter T is the final time instant allotted for the control to reach the final state. The parameter T determines how quickly the system is required to converge. Small values of T will make the system difficult to control (6). In theory, T is dimensionless if not coupled to external time domains. As we do not intend to model an evolving process in real time, we chose $T = 1$ to use a normalized time. We have expanded the respective paragraph and added further references to works providing a more general introduction:

SI page 5:

“The time horizon T specifies the time over which the control input is applied and the system can be pushed from one state to the other. It determines how quickly the system is required to converge and therefore small values might give the system insufficient time to reach the target state, making it hard to control. In theory, T is dimensionless if not coupled to external time domains. As we do not intend to model an evolving process in real time, we chose $T = 1$ to use a normalized time, in line with previous works (12), which allows the system to have adequate time to be controlled (13). For a systematic investigation of the influence of T on control processes in the context of brain networks, we refer the reader to (13), as well as to more general introductions of the control processes in linear dynamics (14, 15).”

The parameter ρ allows the investigator to penalize the energy used by the optimal input in relation to the deviation from the optimal trajectory when solving for the optimal control. As we had no specific hypothesis that either of these elements of the cost function should prevail over the other, we used the default of $\rho = 1$. In the main text, we refer to it as $\tilde{\rho}$, which we have now harmonized. We apologize for the confusion.

The matrix S is the $n \times n$ diagonal matrix with 1 on each diagonal entry for which the distance from the final target state is penalized. Since we were interested in studying the case in which all states reach the final target state, the input here was $S = \text{eye}(n)$ (i.e., a diagonal matrix with all ones on the diagonal), as in previous publications (7).

For the benefit of future readers, we have added a more detailed description of S and T to the README file.

README file:

- S** (NxN) Selects nodes whose distance from the final target state you wish to penalize in the cost function. Define S so that there is a 1 on the diagonal of elements you want to constrain, and a zero otherwise.
- T** Time horizon: how long you allow for the state trajectory to reach the target state as a result of the input injected into the control nodes. Small values of T will make the system difficult to control whereas large values of T will make the system easy to control. In theory, T is dimensionless if not coupled to external time domains; therefore $T = 1$ can be used to have a normalized time.

6. In line 446 it reads that B denotes an input matrix defining the control nodes. Line 116 reads "B is a matrix describing which regions enact control or receive control input".

I find the definition in the methods section more appropriate: How can you distinguish between sending (control) and receiving nodes if not using some measure of effective connectivity?

We thank the reviewer for this comment. Indeed, we do not distinguish between enacting or receiving control nodes here and therefore have changed the description in line with your suggestion. For a discussion of controllability in directed connectomes, please see our earlier paper Kim *et al.* 2018 *Nature Physics*.

other issues:

7. Is the following sentence correct? Are we talking about a diversified energy landscape or a less diversified energy landscape?
"In a diversified energy landscape, we expected that the variation of trajectories around the minimum-energy trajectory should be larger than expected in the less diversified energy landscape in schizophrenia, implying that small perturbations may have a more substantial impact."

We thank the reviewer for pointing out this crucial typo. The passage should indeed read as follows:

"In a diversified energy landscape, we expected that the variation of trajectories around the minimum-energy trajectory should be larger than ~~expected~~ in the less diversified energy landscape of healthy subjects, implying that small perturbations may have a more substantial impact in schizophrenia."

8. Figure S4: Is there any reason that subcortical regions were omitted from the figure?

Subcortical regions do not score among the 50% most important control nodes, which is why we refrained from plotting the subcortical structures in the previous figures. However, for Fig S4 D and E, subcortical structures could be potentially informative, so we now display those results using a separate subcortical template.

Reviewer #3 (Remarks to the Author):

The authors have adequately addressed my concerns.

We thank the reviewer for their appreciation of our revisions.

References

1. Jonas KG, Lencz T, Li K, Malhotra AK, Perlman G, Fochtmann LJ, et al. (2019): Schizophrenia polygenic risk score and 20-year course of illness in psychotic disorders. *Transl Psychiatry*. 9:300.
2. Horsdal HT, Agerbo E, McGrath JJ, Vilhjalmsson BJ, Antonsen S, Closter AM, et al. (2019): Association of Childhood Exposure to Nitrogen Dioxide and Polygenic Risk Score for Schizophrenia With the Risk of Developing Schizophrenia. *JAMA Netw Open*. 2:e1914401.
3. Erk S, Mohnke S, Ripke S, Lett TA, Veer IM, Wackerhagen C, et al. (2017): Functional neuroimaging effects of recently discovered genetic risk loci for schizophrenia and polygenic risk profile in five RDoC subdomains. *Transl Psychiatry*. 7:e997.
4. Walton E, Geisler D, Lee PH, Hass J, Turner JA, Liu J, et al. (2014): Prefrontal inefficiency is associated with polygenic risk for schizophrenia. *Schizophrenia bulletin*. 40:1263-1271.
5. Kailath T (1980): *Linear systems*. Prentice-Hall Englewood Cliffs, NJ.
6. Karrer TM, Kim JZ, Stiso J, Kahn AE, Pasqualetti F, Habel U, et al. (2020): A practical guide to methodological considerations in the controllability of structural brain networks. *J Neural Eng*.
7. Betzel RF, Gu S, Medaglia JD, Pasqualetti F, Bassett DS (2016): Optimally controlling the human connectome: the role of network topology. *Sci Rep*. 6:30770.

REVIEWERS' COMMENTS

Reviewer #1 (Remarks to the Author):

The authors have adequately addressed my comments. I anticipate that the paper will be well received by the community

Reviewer #2 (Remarks to the Author):

I appreciate the authors' detailed responses to my questions. I hope that readers of the published manuscript will find the additional explanations as helpful as I did.

I am impressed by the authors' work and intrigued by the new insights on dual state theory. I have nothing further to ask.

Thanks for allowing me to be part in this review process.

Best wishes

Sebastian Markt

REVIEWER COMMENTS

Reviewer #1 (Remarks to the Author):

The authors have adequately addressed my comments. I anticipate that the paper will be well received by the community

We thank the reviewer for their time and helpful comments along the way, which helped to improve the manuscript significantly.

Reviewer #2 (Remarks to the Author):

I appreciate the authors' detailed responses to my questions. I hope that readers of the published manuscript will find the additional explanations as helpful as I did.

I am impressed by the authors' work and intrigued by the new insights on dual state theory. I have nothing further to ask.

Thanks for allowing me to be part in this review process.

Best wishes

Sebastian Markett

We thank the reviewer for his time and effort in helping us to improve our manuscript significantly. We are especially thankful for his comments on the network control framework, which were essential in providing a thorough discussion and explanation to guide readers through these concepts.